# Gompertz Linear Units: Leveraging Asymmetry for Enhanced Learning Dynamics

**Indrashis Das**
University of Freiburg
dasi@cs.uni-freiburg.de

**Mahmoud Safari**
University of Freiburg
safarim@cs.uni-freiburg.de

**Steven Adriaensen**
University of Freiburg
adriaens@cs.uni-freiburg.de

**Frank Hutter**
Prior Labs & ELLIS Institute Tübingen & University of Freiburg
fh@cs.uni-freiburg.de

## Abstract

Activation functions are fundamental elements of deep learning architectures as they significantly influence training dynamics. ReLU, while widely used, is prone to the dying neuron problem, which has been mitigated by variants such as LeakyReLU, PReLU, and ELU that better handle negative neuron outputs. Recently, self-gated activations like GELU and Swish have emerged as state-of-the-art alternatives, leveraging their smoothness to ensure stable gradient flow and prevent neuron inactivity. In this work, we introduce the Gompertz Linear Unit (GoLU), a novel self-gated activation function defined as $\mathrm{GoLU}(x) = x\,\mathrm{Gompertz}(x)$, where $\mathrm{Gompertz}(x) = e^{-e^{-x}}$. The GoLU activation leverages the right-skewed asymmetry in the Gompertz function to reduce variance in the latent space more effectively compared to GELU and Swish, while preserving robust gradient flow. Extensive experiments across diverse tasks, including Image Classification, Language Modeling, Semantic Segmentation, Object Detection, Instance Segmentation, and Diffusion, highlight GoLU's superior performance relative to state-of-the-art activation functions, establishing GoLU as a robust alternative to existing activation functions.

## 1 Introduction

Developing effective activation functions has been a longstanding area of research in deep learning. In the early days, the Sigmoid [Verhulst, 1838, Rumelhart et al., 1986] and Tanh [LeCun et al., 2002] functions were popular choices. However, these activations can suffer from the vanishing gradient problem due to their tendency to saturate. The introduction of ReLU [Nair and Hinton, 2010] marked a turning point, as it allowed for more efficient training by alleviating the vanishing gradient problem and inducing intensity equivariance [Nair and Hinton, 2010]. However, ReLU comes with its own challenges, notably the dying-ReLU problem. To address these challenges, several ReLU variants have been developed, including LeakyReLU [Maas et al., 2013], PReLU [He et al., 2015] and ELU [Clevert et al., 2015]. Despite the emergence of these alternatives, ReLU remains one of the most widely used activation functions today, owing to its simplicity as a piecewise linear function and its computational efficiency.

39th Conference on Neural Information Processing Systems (NeurIPS 2025).

In the deep learning community, the landscape of activation functions has gradually shifted towards self-gated activations such as Gaussian Error Linear Units (GELU) [Hendrycks and Gimpel, 2016], Swish [Ramachandran et al., 2017], and Mish [Misra, 2019]. These activations provide probabilistic interpretations while enhancing robustness when combined with normalization techniques [Ioffe and Szegedy, 2015, Ba et al., 2016, Ulyanov et al., 2016, Wu and He, 2018, Zhang and Sennrich, 2019]. Unlike ReLU, which strictly enforces gradient preservation due to its piecewise-linear nature, Swish, Mish and GELU, as smooth activation functions, relax these constraints. Their smoothness allows for improved gradient flow without strictly adhering to intensity equivariance.

In this work we introduce Gompertz Linear Units (GoLU), a new activation function of the self-gated family based on the Gompertz function [Gompertz, 1825] as its gating mechanism. The Gompertz function was initially developed to model human mortality rates, and has since been widely applied in biology. Notably, it also possesses a probabilistic interpretation, as it represents the cumulative distribution function (CDF) of the standard Gumbel distribution. While both the Sigmoid function and the Gaussian CDF exhibit reflection symmetry around the point (0, 0.5), the Gompertz function manifests a subtle rightward asymmetry, leading to distinct qualitative behavior.

Our experiments indicate that GoLU, compared to existing self-gated activations, effectively *reduces variance* in the latent representation. Moreover, it contributes to a *smoother loss landscape*, making it less sensitive to small perturbations in the model parameters. Additionally, an analysis of the learned weights in our trained models reveals that GoLU induces a more *spread weight distribution* compared to commonly used activations (see Section 2.2 for details).

A more spread weight distribution may indicate the network's ability to capture a diverse range of features from the data. On the other hand, variance reduction in activation outputs can help eliminate irrelevant information, allowing the network to focus on distinguishing patterns and potentially mitigate overfitting. However, overly broad weight distributions may introduce instability, while excessive variance reduction could result in underfitting and the loss of essential features, ultimately degrading performance.

Extensive, task-specific evaluations, suggest that GoLU effectively addresses this trade-off by achieving a balanced level of both weight distribution and variance reduction, leading to improved performance over baseline activations (see Section 3). To facilitate reproducibility, we have made our code available at `https://github.com/automl/GoLU`.

## 2 Gompertz Linear Unit

### 2.1 Definition and Properties

In this section, we introduce the GoLU activation function and discuss its properties. GoLU is defined through Equations 1 and 2 and visualized in Figure 1 (Left) as the red curve, alongside other activation functions for comparison.

$$\text{GoLU}(x) = x \, \text{Gompertz}(x) \tag{1}$$

$$\text{Gompertz(x)} = e^{-e^{-x}} \tag{2}$$

The gate function $\text{Gompertz}(x)$ refers to the Gompertz function introduced in [Gompertz, 1825] and is plotted in red in Figure 1 (Right), alongside the gate functions of other gated activations. The Gompertz function can also be interpreted probabilistically, as it corresponds to the CDF of the standard Gumbel distribution, $\text{Gumbel}(0, 1)$, with probability density function

$$\text{Gumbel}(x) = e^{-(x+e^{-x})} \tag{3}$$

From Equations 1, 2 and Figure 1, we understand that, contrary to ReLU and its variants which are monotonic and non-smooth at zero, GoLU is a smooth and non-monotonic self-gated activation, similar to Swish and GELU. In fact the formulation of GoLU using exponentials makes it infinitely differentiable. However, in contrast to Sigmoid and the Gaussian CDF (i.e. the gate functions of Swish and GELU), the Gompertz function is asymmetric, as it does not mirror evenly around a central point[1].

---

[1] Formally, we refer to a scalar function $f(x)$ as symmetric if there exists a point $x^*$ such that for any input $x$ we have $f(x^* + x) - f(x^*) = f(x^*) - f(x^* - x)$.

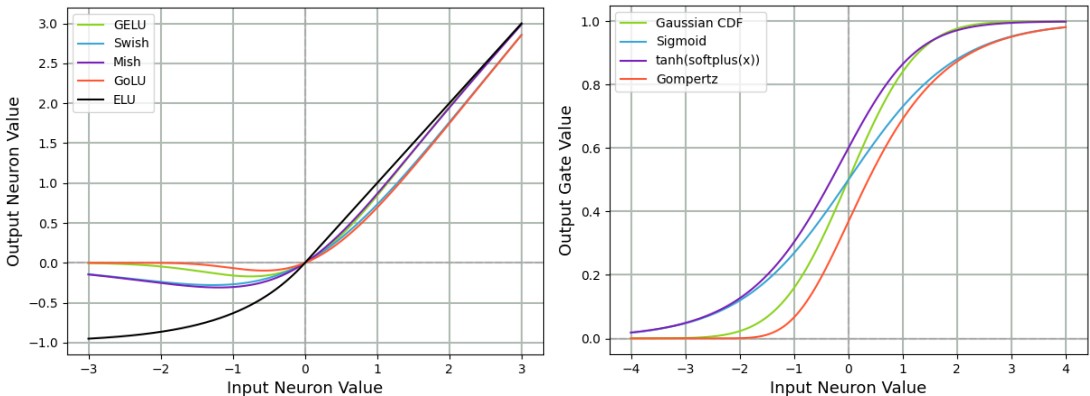

Figure 1: Activation functions (Left) and their corresponding gates (Right). GoLU and its gate, the Gompertz function, are highlighted in red. Note the slight rightward shift of the Gompertz gate.

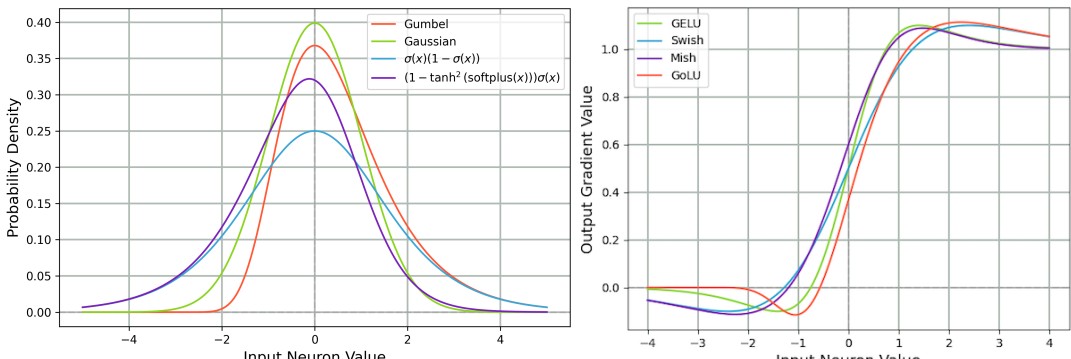

Figure 2: Comparison of the distributions underlying the gate functions (Left) and the gradients (Right) of various gated activations. The Gumbel distribution exhibits a slight rightward skew.

This asymmetry, which has a bias towards the right, arises from the inherent asymmetry of the Gumbel distribution, which favors positive input values, as illustrated in Figure 2 (Left). In fact, the right-leaning asymmetry of the Gumbel distribution leads to smaller gate values across the entire input range, inducing a compression effect on the output distribution. This behavior extends to GoLU, yielding output values closer to zero, both for positive and negative inputs, when compared to other gated activation functions, effectively reducing the magnitude of the activation output. We note that, while Mish also exhibits an asymmetric distribution, it is skewed to the left, producing the opposite effect relative to GoLU[2].

From a more localized perspective, the Gompertz gate exhibits a reduced value in particular at the origin. This leads to a decreased steepness of GoLU near this point, as indicated by $\text{GoLU}'(0) = \text{Gompertz}(0)$ from Equation 1. This property of reduced slope magnitude is not confined to the origin but extends to a neighborhood around it and spans a substantial portion of the negative input domain, as shown in Figure 2 (Right). Additional details are provided in Appendix A.

In the large negative region, the Gompertz gate, and consequently the GoLU activation, decays extremely rapidly as a double exponential, suppressing unimportant features like ReLU, while maintaining smoothness, unlike ReLU.

Compared to the Gaussian CDF and the Sigmoid function, the Gompertz gate initially exhibits a flat plateau, followed by a steeper growth rate that aligns more closely with the Gaussian CDF. As the input values become large and positive, the growth rate flattens and resembles the Sigmoid function, with the difference falling off as $\mathcal{O}(e^{-2x})$ (see Appendix A).

---

[2]See Appendix B for an interesting case of flipped Mish distribution with right-leaning asymmetry.

## 2.2 Effects on Training Dynamics

The distinctive properties of GoLU influence the training dynamics, as we will outline here.

**Variance reduction**    As illustrated in Figure 1 (Left), GoLU exhibits a profile that remains closest to the x-axis across the entire input range. Moreover, its slope, particularly near the origin and over a substantial portion of the negative input domain, is smaller in magnitude compared to other gated activations, as pointed out in Section 2.1. These characteristics suggest a reduced sensitivity of the activation output to variations in the input. In fact, for a scalar activation function $f$ the variance of its output can be shown to be approximately proportional to the square of its slope

$$\text{Var}[f(x)] \approx f'(\mu)^2 \sigma^2 \tag{4}$$

where $\mu$ and $\sigma^2$ denote the mean and variance of the input, respectively (see Appendix A for the derivation). This analytic relation explains more directly how the smaller slope of GoLU contributes to a lower variance in its output. As a result, GoLU effectively reduces variance in the latent representations, and promotes smoother activation outputs, enhancing the model's ability to distinguish between strong and weak features.

To visually illustrate this phenomenon, we process Figure 3 (Left) through a $3 \times 3$ 2D Convolution followed by 2D Batch Normalization. The resulting pre-activation is then passed through various activation functions, and the pixel distributions of the normalized pre-activation and activation maps are plotted for GoLU, GELU, and Swish in Figure 3 (Right). As observed, GoLU exhibits a distinctive "squeezing effect", compressing the same distribution into a smaller output range, and reducing variance most, compared to GELU and Swish.

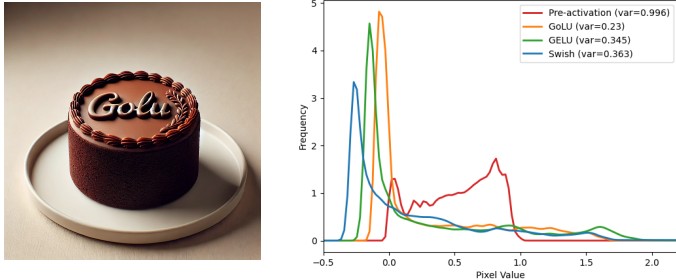

Figure 3: Image created by Dall-E 3 (Left) and kernel density estimation curves for distributions of activation outputs for the image (Right). GoLU reduces variance most compared to baseline activations.

To further substantiate this observation, we randomly sample four images from the CIFAR-10 dataset, apply the same preprocessing pipeline, and pass the results through different activation functions. The variances of the activated signals, summarized in Table 1, highlight GoLU's ability to achieve a notable reduction in variance compared to widely-used activations, enabling smoother data representation. This effect is not limited to CIFAR-10, as we also observe a similar trend on ImageNet. Finally, to illustrate this effect in a fully trained model, we randomly sample three images from the ImageNet-1k dataset and pass them through a ResNet-50 model trained on ImageNet-1k. As shown in Figure 4, the output distributions of the final activations demonstrate that GoLU produces a more peaked distribution compared to other activation functions, highlighting this distinctive effect on latent representations. We note that this effect is not limited to the final activation, as shown in Figure 4, but is consistently observed across intermediate layers of ResNet-50 as well. This lower activation variance can be seen as a form of implicit regularization as the network's representation of the input becomes smoother, focusing on the core patterns rather than fine-grained details or noise.

**Smooth loss landscape**    Reduced activation variance results in less noisy and more consistent gradients. This typically means that the loss function changes more smoothly with respect to model parameters. As a result, the optimizer is more likely to converge to flatter regions of the loss landscape with smaller curvature. This is expected to result in better robustness to small perturbations of the

Table 1: Variances of randomly sampled images from CIFAR-10 after applying a 3x3 Convolution followed by Batch Normalization and further passing the feature maps through different activations.

| Activation | 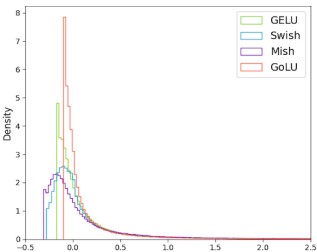 | | | |
|:---:|:---:|:---:|:---:|:---:|
| ReLU | 0.3024 | 0.3063 | 0.3627 | 0.3594 |
| LeakyReLU | 0.3055 | 0.3100 | 0.3639 | 0.3626 |
| ELU | 0.5677 | 0.6227 | 0.4699 | 0.6418 |
| GELU | 0.2995 | 0.3102 | 0.3583 | 0.3701 |
| Swish | 0.2685 | 0.2872 | 0.3332 | 0.3399 |
| Mish | 0.3448 | 0.3700 | 0.3677 | 0.4200 |
| **GoLU** | **0.2133** | **0.2150** | **0.3213** | **0.2783** |

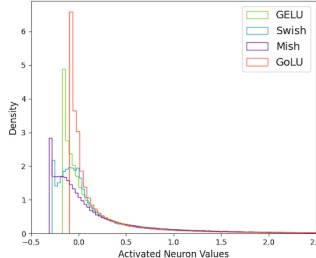
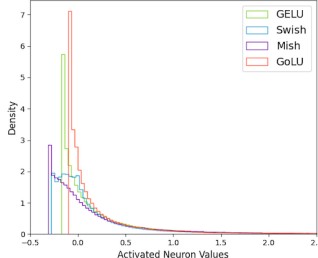

Figure 4: Distributions of final activation outputs of ResNet-50 trained on ImageNet-1k for three randomly sampled images from ImageNet-1k. GoLU leads to a more peaked distribution for the final activation output.

model parameters. We explore this by adding two different Standard Normal noise terms, scaled independently by $\alpha, \beta$, to the weights of ResNet-20 trained on CIFAR-10. We compute the test loss across a grid of scaling factors $\alpha, \beta$ for the two terms, while keeping the noises constant (refer to Appendix C for more details). ResNet-20 with GoLU shows relatively smoother, less-spiked loss landscapes compared to other activations (Figure 5) which is consistent with GoLU's better generalization gains reported in the next section. Additionally, GoLU exhibits a lower average loss, as shown in Figure 5. We note that Figure 5 offers qualitative insight into the smoothness of the loss landscape along a 2D random subspace and is not intended as a rigorous or quantitative analysis. Figure 9 in Appendix C presents a comparison of loss value distributions across the loss landscape, indicating that GoLU yields lower loss variance compared to other activation functions.

**Spread weight distribution**    In contrast to the reduced variance in the latent space, we observe a wider distribution in the learned weights of our models trained with GoLU, at least in the region where most weights are concentrated. Figure 6 compares non-normalization[3] weight distributions of ResNet-50 and ViT-B/32 trained on ImageNet-1k and GPT2-S (124M) trained on OpenWebText, with different activation functions. The broader weight distribution for GoLU around the peak suggests that the network has learned more diverse transformations, enhancing its capacity to distinguish between features in the data.

This may reflect the network's response to reduced activation variance, counterbalancing it by spreading the weights around the peak to maintain representational diversity. Specifically, reduced output variance naturally leads to more uniform gradients, which in turn encourages a broader spread of weights.

Notice that a wider weight distribution around the peak does not necessarily translate to a larger *overall* variance. However, focusing on the bulk of the distribution[4], we find that GoLU consistently achieves the highest variance. This behavior suggests that networks trained with GoLU effectively

---

[3]As learned transformations in the model are mainly encoded in the weights of fully connected, convolutional or attention layers, it is more meaningful to exclude parameters of Batch Normalization and Layer Normalization layers, although including these parameters we obtain qualitatively similar distributions.

[4]Specifically, we take the intersection of the middle 98% intervals of the parameter distributions of an architecture trained with each activation.

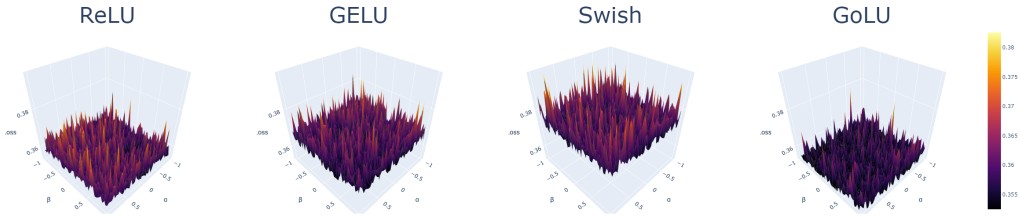

Figure 5: The loss landscape on the test set of ResNet-20 trained on CIFAR-10 with ReLU, GELU, Swish and GoLU after adding random, scaled perturbations to the learned weights (refer to Appendix C for more details).

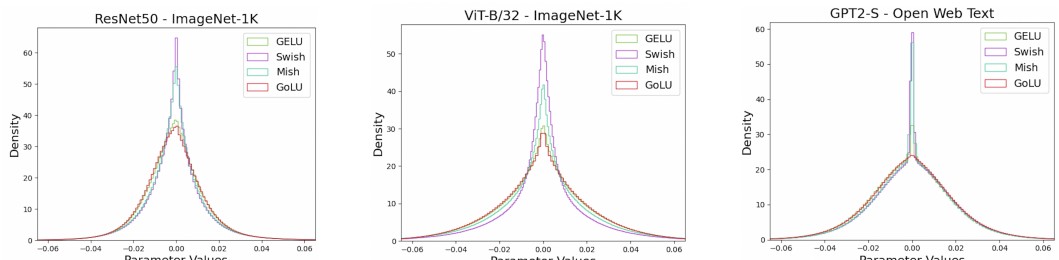

Figure 6: Learned-weight distribution of ResNet-50 and ViT- B/16 trained on ImageNet-1k and GPT2-S trained on OWT. GoLU leads to a more spread weight distribution. The range of parameters is clipped for better visualization.

suppress density in extreme values while expanding the distribution around the peak. Such a pattern implies that the model captures a broader range of *meaningful* transformations without over-reliance on extreme parameter values or certain features.

We emphasize that the effects attributed to GoLU, as described above, are not guaranteed to hold universally across all scenarios but rather represent general trends observed in our empirical findings.

Moreover, while asymmetry has been highlighted as a distinctive feature of GoLU, it is important to note that its high performance, detailed in the next section, cannot be solely attributed to asymmetry, but arises from an intricate interplay of properties, described in Section 2.1.

## 3 Experiments and Results

### 3.1 Overview of Experiments

We conducted experiments across various architectures and datasets, spanning a diverse range of tasks in both vision and language modeling. We begin with image classification, training ResNet-18, 34, 50 [He et al., 2016], WideResNet-50-2 [Zagoruyko and Komodakis, 2016], DenseNet-121 [Huang et al., 2017], EfficientNet-B0 [Tan and Le, 2019], TinyViT Wu et al. [2022], ViT-B/32 and ViT-B/16 [Dosovitskiy et al., 2020] on ImageNet-1k Deng et al. [2009].

We then extend our experiments to language modeling. We train babyGPT on the TinyStories (TS) [Eldan and Li, 2023] dataset and GPT2-S [Radford et al., 2019] on the OpenWebText (OWT) [Gokaslan et al., 2019] dataset, leveraging the nanoGPT repository Karpathy [2023].

Additionally, we assess GoLU's performance on Semantic Segmentation (DeepLabV3 Chen et al. [2017]), Object Detection (Faster R-CNN-FPN Ren et al. [2015], RetinaNet-FPN Lin [2017]), and Instance Segmentation (Mask R-CNN-FPN He et al. [2017]) on MS-COCO Lin et al. [2014], leveraging our pre-trained ResNet-50 backbone on ImageNet-1k. Further, we test GoLU on Denoising Diffusion Probabilistic Models Ho et al. [2020] on the CelebA Liu et al. [2015] dataset.

We closely follow established baselines for all model architectures and tasks, ensuring that the integration of GoLU is the primary change. Hyperparameters, optimizers, learning rate schedules,

Table 2: Top-1 test accuracy of ResNets 18, 34 and 50, WideResNet-50-2, DenseNet-121, EfficientNet-B0, TinyViT, ViT-B/32 and ViT-B/16 on ImageNet-1k.

| Architecture | ReLU | LeakyReLU | ELU | GELU | Swish | Mish | GoLU |
|---|---|---|---|---|---|---|---|
| RN-18 | 69.74±0.07 | 69.78±0.04 | 67.10±0.07 | 70.66±0.05 | 70.60±0.06 | 70.53±0.06 | **70.76±0.06** |
| RN-34 | 73.26±0.01 | 73.25±0.03 | 69.27±0.09 | 73.44±0.04 | 72.74±0.05 | 72.73±0.07 | **73.71±0.04** |
| RN-50 | 75.44±0.07 | 75.67±0.08 | 71.87±0.09 | 76.07±0.06 | 75.17±0.14 | 75.53±0.09 | **76.63±0.03** |
| WRN-50-2 | 76.96±0.07 | 77.17±0.12 | 71.90±0.01 | 76.72±0.01 | 75.41±0.03 | 75.75±0.19 | **77.37±0.03** |
| DN-121 | 74.95±0.09 | 75.03±0.06 | 68.95±0.04 | 74.64±0.11 | 72.81±0.06 | 72.97±0.10 | **75.25±0.03** |
| EN-B0 | 76.52±0.07 | 76.65±0.04 | 76.21±0.04 | **76.90±0.01** | 76.84±0.02 | 76.76±0.06 | 76.86±0.04 |
| TinyViT | 82.91±0.02 | 82.83±0.03 | 80.29±0.07 | 83.05±0.03 | 82.92±0.06 | 83.01± 0.02 | **83.21±0.02** |
| ViT-B/32 | 74.51±0.04 | 74.53±0.03 | 65.82±0.07 | 75.48±0.05 | 72.31±2.15 | 75.16±0.07 | **75.74±0.09** |
| ViT-B/16 | 80.06±0.05 | 79.93±0.02 | 73.36±0.16 | 79.39±0.99 | 79.19±0.94 | 77.97±1.95 | **80.72±0.04** |

and other training settings are aligned with the standard practices for each task. All our experiments are conducted on three seeds and the results are averaged out and reported with the standard error.

In Appendix E we further present a Critical Difference analysis to systematically compare the overall performance of activation functions. In Appendix I, we assess the performance of GoLU on a machine translation task, using the WMT14 English–German benchmark. Finally, in Appendix J, we explore the application of GoLU to the task of learning curve extrapolation.

## 3.2 Image Classification

We evaluate GoLU's performance in image classification tasks on ImageNet-1k, comparing it against six state-of-the-art activation functions, ReLU, LeakyReLU, ELU, GELU, Swish and Mish. Table 2 presents the top-1 test accuracies with standard errors for ResNets 18, 34 and 50, WideResNet-50-2, DenseNet-121, EfficientNet-B0, ViT-B/32, ViT-B/16 and TinyViT Wu et al. [2022]. The training settings, detailed in Appendix G.1, are adopted from Torchvision TorchVision [2016] for all experiments except EfficientNet-B0 which is taken from the timm library Wightman [2019] and TinyViT which is taken from Wu et al. [2022]. As highlighted, GoLU consistently outperforms the standard activation functions across all architectures, with the exception of EfficientNet-B0, where the performance difference is minimal. Notice that EfficientNet-B0 is an exception because its nonlinearity arises not only from activation functions (which are replaced) but also from a squeeze-and-excitation block, which remains unchanged in our experiments. For ResNet-50 and ViT-B/32, test loss and test accuracy curves are shown in Figures 17 and 18, respectively, where GoLU consistently delivers lower test loss and higher top-1 accuracy over the epochs. GELU is generally the second-best performer, while ELU performs worst across most architectures.

We further evaluate GoLU on CIFAR-10, comparing it against top baseline activations. We report in Table 3 the results of image classification on CIFAR-10, with ResNets 20, 32, 44, 56, and 110, WideResNet28-2, DenseNet40 and ViT-Ti/16-224. GoLU consistently outperforms the standard baselines across all tested architectures. We have further underlined the second-best activations for each model. No single activation consistently ranks second.

Table 3: Top-1 test accuracy on CIFAR-10. GoLU consistently outperforms baselines. Second best activations are underlined.

| Arch. | ReLU | LeakyReLU | GELU | Swish | GoLU |
|---|---|---|---|---|---|
| RN-20 | 91.41±0.1 | 91.60±0.0 | 91.62±0.0 | 91.64±0.1 | **91.77±0.1** |
| RN-32 | 92.21±0.1 | 92.40±0.0 | 92.54±0.1 | 92.16±0.0 | **92.69±0.1** |
| RN-44 | 92.58±0.0 | 92.78±0.0 | 92.78±0.8 | 92.51±0.0 | **92.85±0.0** |
| RN-56 | 92.80±0.1 | 92.75±0.1 | 92.86±0.1 | 92.93±0.1 | **93.15±0.1** |
| RN-110 | 93.21±0.0 | 93.18±0.1 | 92.75±0.1 | 92.23±0.0 | **93.25±0.0** |
| WRN-28-2 | 94.96±0.0 | 94.81±0.0 | 94.55±0.1 | 93.58±0.1 | **95.03±0.0** |
| DN-40 | 93.13±0.1 | 93.13±0.1 | 93.41±0.0 | 93.30±0.1 | **93.44±0.1** |
| ViT-Ti | 91.74±0.06 | 91.61±0.18 | 91.37±0.11 | 88.61±0.16 | **92.60±0.05** |

Table 4: Test perplexity score and test token accuracy of babyGPT and GPT2-S trained on TS and OWT respectively.

| Activation | babyGPT - TinyStories | | GPT2-S - OpenWebText | |
|---|---|---|---|---|
| | Perplexity | Token Accuracy | Perplexity | Token Accuracy |
| ReLU | 4.519±0.006 | 61.243±0.030 | 17.845±0.078 | 44.059±0.079 |
| LeakyReLU | 4.516±0.005 | 61.237±0.032 | 17.778±0.125 | 44.103±0.074 |
| ELU | 4.872±0.005 | 59.859±0.027 | 18.375±0.035 | 43.721±0.040 |
| GELU | 4.462±0.005 | 61.465±0.034 | 17.525±0.015 | 44.262±0.042 |
| Swish | 4.535±0.004 | 61.178±0.032 | 17.785±0.026 | 44.155±0.025 |
| Mish | 4.539±0.007 | 61.135±0.036 | 17.797±0.086 | 44.104±0.081 |
| GoLU | **4.444±0.005** | **61.545±0.029** | **17.297±0.023** | **44.413±0.023** |

Table 5: Test loss and test mIoU of DeepLabV3 ResNet-50 trained on MS-COCO.

| Activation | LR=0.02 | | LR=0.01 | |
|---|---|---|---|---|
| | Test Loss | Test mIoU | Test Loss | Test mIoU |
| ReLU | 0.344±0.003 | **64.99±0.173** | 0.350±0.004 | 65.11±0.326 |
| LeakyReLU | 0.342±0.003 | 64.79±0.122 | 0.350±0.002 | 65.55±0.131 |
| ELU | 0.367±0.001 | 59.31±0.065 | 0.358±0.001 | 60.70±0.089 |
| GELU | 0.341±0.002 | 64.53±0.136 | **0.341±0.003** | 65.59±0.162 |
| Swish | 0.348±0.003 | 62.52±0.034 | 0.345±0.002 | 64.14±0.135 |
| Mish | 0.344±0.001 | 62.97±0.022 | 0.342±0.002 | 64.40±0.144 |
| GoLU | **0.339±0.000** | 64.98±0.129 | 0.341±0.001 | **65.98±0.124** |

Table 6: Test Box mAP of Faster R-CNN-FPN ResNet-50 and RetinaNet-FPN ResNet-50 trained on MS-COCO.

| Activation | Box mAP | |
|---|---|---|
| | Faster R-CNN | RetinaNet |
| ReLU | 37.44±0.146 | 39.90±0.063 |
| LeakyReLU | 37.41±0.140 | 39.87±0.041 |
| ELU | 35.36±0.041 | 37.43±0.041 |
| GELU | 38.16±0.044 | 40.68±0.090 |
| Swish | 37.28±0.078 | 40.27±0.087 |
| Mish | 37.71±0.087 | 40.45±0.093 |
| GoLU | **38.31±0.058** | **40.77±0.065** |

## 3.3 Language Modeling

We train babyGPT on TS and GPT2-S (124M) on OWT, both sourced from the nanoGPT repository Karpathy [2023]. As shown in Table 4, GoLU demonstrates superior performance, achieving lower perplexity and higher token accuracy on both babyGPT and GPT2-S. GoLU's superiority is also evident in the test loss curves in Figures 19 and 20. The general trend of GELU being the second-best activation function holds in language modeling as well. Appendix G.3 outlines the architectural details and provides additional information on the datasets and training settings.

## 3.4 Semantic Segmentation

For Semantic Segmentation, we train DeepLabV3 on the MS-COCO dataset with PASCAL-VOC labels, from the Torchvision benchmark (see Appendix G.4). We employ our ResNet-50 backbone, pre-trained on ImageNet-1k.

Table 5 (Left) presents the test loss and test mIoU using the original learning rate of 0.02. GoLU achieves the lowest test loss, whereas ReLU attains the highest mIoU, with GoLU ranking second. However, the difference in mIoU between ReLU and GoLU is statistically insignificant.

We conduct a small ablation study on the learning rate and find that lr=0.02 is suboptimal for training the model. Instead, lr=0.01 yields the best performance across all activation functions (see heatmap 10 in Appendix D for full results). Table 5 (Right) reports the results with lr=0.01, where GoLU consistently outperforms other activation functions in terms of mIoU. Additionally, the inference loss and test mIoU curves over epochs, shown in Figures 21 and 22, further emphasize GoLU's strong performance in semantic segmentation.

## 3.5 Object Detection

For Object Detection, we train Faster R-CNN-FPN and RetinaNet-FPN on the MS-COCO dataset. As shown in Table 6 and Figure 23, GoLU outperforms all activation functions on object detection as well, with higher Box mAP (AP @ IoU=0.50:0.95, area=all, maxDets=100) across both Faster R-CNN-FPN and RetinaNet-FPN architectures, while GELU ranks second. Appendix G.5 outlines experimental details.

Table 7: Test Box mAP and Mask mAP of Mask R-CNN-FPN ResNet-50 trained on MS-COCO.

| Activation | LR=0.02 | | LR=0.03 | |
|---|---|---|---|---|
| | Box mAP | Mask mAP | Box mAP | Mask mAP |
| ReLU | 38.33±0.133 | 34.19±0.129 | 38.25±0.144 | 34.28±0.116 |
| LeakyReLU | 38.31±0.230 | 34.19±0.160 | 38.42±0.105 | 34.26±0.065 |
| ELU | 36.41±0.087 | 32.81±0.082 | 35.85±0.069 | 32.38±0.043 |
| GELU | **39.00±0.073** | **34.73±0.024** | 39.12±0.028 | 34.93±0.061 |
| Swish | 38.19±0.224 | 33.99±0.168 | 38.10±0.187 | 33.95±0.112 |
| Mish | 38.76±0.036 | 34.70±0.026 | 38.56±0.138 | 34.64±0.078 |
| GoLU | 38.96±0.101 | 34.54±0.083 | **39.36±0.192** | **34.97±0.146** |

Table 8: Test Loss at LR=0.0003 and LR=0.001 of Denoising Diffusion Probabilistic Model trained on CelebA.

| Activation | Test Loss | |
|---|---|---|
| | LR=0.0003 | LR=0.001 |
| ReLU | 0.0200255±0.0 | 0.0192820±0 |
| LeakyReLU | 0.0200307±0.0 | 0.0192812±0 |
| ELU | 0.0200398±0.0 | 0.0193941±0 |
| GELU | 0.0196956±0.0 | 0.0190221±0 |
| Swish | **0.0196364±0.0** | 0.0190055±0 |
| Mish | 0.0196865±0.0 | 0.0190657±0 |
| GoLU | 0.0196419±0.0 | **0.0189506±0** |

## 3.6 Instance Segmentation

For Instance Segmentation, we train Mask R-CNN-FPN with a ResNet-50 backbone from the Torchvision benchmark on the MS-COCO dataset (see Appendix G.6 for training settings). As shown in Table 7 (Left), GELU achieves the best performance in this setting (with the default lr=0.02), with GoLU ranking second in Box mAP and third in Mask mAP (both implying AP @ IoU=0.50:0.95, area=all, maxDets=100). However, Figure 24, which depicts test Box mAP and Mask mAP over epochs, reveals that GoLU generally outperforms GELU and ReLU throughout the training process. We perform a learning rate ablation study and observe that, similar to the Semantic Segmentation task, a learning rate of 0.02 is suboptimal for this specific architecture–dataset combination. In contrast, increasing the learning rate to 0.03 leads to improved performance across all activation functions (see heatmaps 12 and 13). Surprisingly, at the optimal learning rate of 0.03, GoLU outperforms all baseline activations, as shown in Table 7 (right).

## 3.7 Denoising Diffusion Probabilistic Models

We train a Denoising Diffusion Probabilistic Model on the CelebA dataset (see Appendix G.7). As shown in Table 8, for the default lr=0.0003, gated activations perform comparably to the baseline activation, Swish, which achieves the best performance, with GoLU ranking a close second. Figure 25 (Left) further illustrates the test loss over epochs. Similar to our findings in semantic segmentation and instance segmentation, we conduct a learning rate ablation study. Results, summarized in heatmap 11 in Appendix D, indicate that increasing the lr from the default value of 0.0003 to 0.0004, 0.0005 and 0.001 progressively improves performance across all activations. Notably, for lr values of 0.0004, 0.0005 and 0.001, GoLU achieves the lowest final test loss. Results for the optimum lr=0.001 are highlighted in the right column of Table 8 and Figure 25 (Right). These findings are in line with the trend observed in semantic segmentation and instance segmentation, where GoLU outperforms baseline activations under optimal lr configurations.

# 4 Training and Inference Speed

Existing activation functions in PyTorch leverage CUDA kernels in Eager mode to achieve optimal speedup. To ensure a fair comparison of training and inference speeds, we developed a CUDA-optimized kernel for GoLU, which was used for all training experiments described in the previous sections. Table 9 in Appendix F presents the relative training and inference speeds of GoLU compared to the default activation function across various tasks.

Our results show that GoLU achieves a speed comparable to that of the default activation function across all architectures. The only exception is DeepLabV3-ResNet-50 trained on MS-COCO, where GoLU incurs slightly higher training time. However, this is consistent with other activation functions, all of which exhibit increased training times relative to ReLU in this specific architecture.

# 5 Conclusions

We have introduced GoLU, a new self-gated activation function based on the CDF of the Gumbel distribution as its gate function. Through extensive analysis and experiments, we have demonstrated

that GoLU provides a regularising effect by reducing variance in the activation output, it enables the representation of diverse features through a more distributed weight pattern, and encourages a smoother and more robust loss landscape. Notably, our results show that GoLU generally outperforms state-of-the-art baseline activation functions across a wide range of tasks and domains, from computer vision to language modeling. Additionally, we implemented a custom CUDA kernel to optimize training and inference efficiency, minimizing latency and enhancing scalability. GoLU offers a robust, efficient, and scalable alternative to existing activation functions. Its integration into state-of-the-art neural networks has the potential to improve performance across various applications, positioning GoLU as a promising standard in modern deep learning.

## Acknowlwdgements

This research was funded by the Deutsche Forschungsgemeinschaft (DFG, German Research Foundation) under grant number 539134284, through EFRE (FEIH_2698644) and the state of Baden-Württemberg.

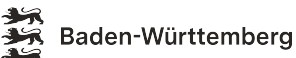 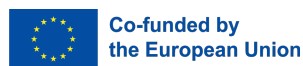

The authors gratefully acknowledge the computing time made available to them on the high-performance computer NHR@KIT Compute Cluster at the NHR Center NHR@KIT. These Centers are jointly supported by the Federal Ministry of Education and Research and the state governments participating in the NHR (www.nhr-verein.de/unsere-partner). We acknowledge funding by the European Union (via ERC Consolidator Grant DeepLearning 2.0, grant no. 101045765). Views and opinions expressed are however those of the author(s) only and do not necessarily reflect those of the European Union or the European Research Council. Neither the European Union nor the granting authority can be held responsible for them.



The authors gratefully acknowledge the Gauss Centre for Supercomputing e.V. (www.gauss-centre.eu) for funding this project by providing computing time on the GCS Supercomputer JUWELS at Jülich Supercomputing Center (JSC) and SuperMUC-NG at Leibniz Supercomputing Centre (www.lrz.de). The authors acknowledge support by the state of Baden-Württemberg through bwHPC. We acknowledge the EuroHPC Joint Undertaking for awarding this project access to the EuroHPC supercomputer LEONARDO, hosted by CINECA (Italy) and the LEONARDO consortium through an EuroHPC Extreme Access grant EHPC-EXT-2023E02-068. Frank Hutter acknowledges the financial support of the Hector Foundation. We also thank Jörg Franke, Edward Bergman, Arbër Zela, André Biedenkapp and Lennart Purucker for their constructive feedback throughout the development of this work.

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

# A  Properties of GoLU: Further Details

To further elucidate the concepts presented in Section 2.1 and gain deeper insights into the properties of GoLU, we present additional details and visualizations in this section.

Figure 7 compares the GoLU activation with GELU, highlighting how the right-leaning inclination of the Gumbel distribution, in contrast to the symmetric Gaussian distribution (Left column), results in a smaller value of the Gompertz gate at the origin compared to the Gaussian CDF (Middle column). In fact, this behavior is not confined to the origin, and the Gompertz gate remains smaller than the Gaussian CDF across the entire input range.

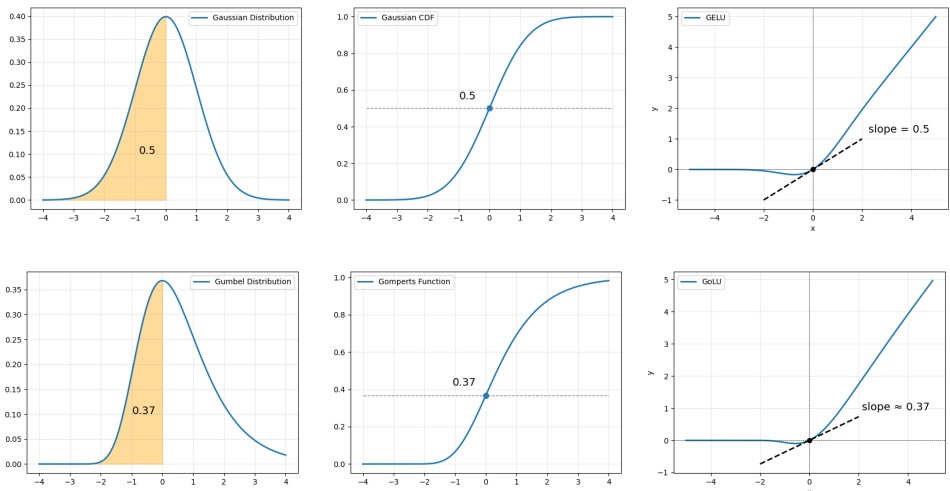

Figure 7: Top row, from left to right: Gaussian distribution, Gaussian CDF, GELU. Bottom row, from left to right: Gumbel distribution, Gompertz function, GoLU.

This reduced value of the Gompertz gate at the origin directly translates into a lower slope for GoLU compared to GELU, as illustrated in Figure 7 (Right column). This can be readily seen by taking the derivative of the GoLU activation and evaluating it at zero

$$\text{GoLU}'(x) = x\,\text{Gompertz}'(x) + \text{Gompertz}(x) \tag{5}$$
$$\text{GoLU}'(0) = \text{Gompertz}(0) \tag{6}$$

which shows that the slope of GoLU at the origin corresponds to the value of the Gompertz gate at the origin. Similarly, the slope of GELU at the origin is determined by the Gaussian CDF at the origin.

Assuming the input distribution resembles a zero-centered, nearly-Gaussian form, which is likely particularly when employing batch normalization and appropriate weight initialization, the activations can be approximated by their tangents at the origin. Therefore a reduced slope at the origin translates into decreased sensitivity to input variations and lower output variance. We note that GoLU exhibits a lower slope magnitude not only in a neighborhood around the origin but across a significant portion of the negative input domain as illustrated in Figure 2.

More generally, it can be shown analytically that for a given activation function $f$ that is smooth in a neighborhood of its input mean, and for a sufficiently localized input distribution, the variance of the activation output is approximately proportional to the square of its slope evaluated at the mean input $\mu$, with the input variance $\sigma^2$ serving as the proportionality constant

$$\text{Var}[f(x)] \approx f'(\mu)^2\sigma^2 \tag{7}$$

This formally demonstrates how smaller activation slopes result in reduced output variance.

To derive this connection, we apply the definition of variance to a scalar activation function $f(x)$:

$$\text{Var}[f(x)] = \mathbb{E}[(f(x) - \mathbb{E}[f(x)])^2] = \mathbb{E}[f(x)^2] - \mathbb{E}[f(x)]^2 \tag{8}$$

Expanding $f(x)$ and $f(x)^2$ in a Taylor series around the mean input $\mu$, gives:

$$f(x) = f(\mu) + f'(\mu)(x - \mu) + \frac{1}{2}f''(\mu)(x - \mu)^2 + \cdots \tag{9}$$

$$f(x)^2 = f(\mu)^2 + 2f(\mu)f'(\mu)(x - \mu) + f'(\mu)^2(x - \mu)^2 + f(\mu)f''(\mu)(x - \mu)^2 + \cdots \tag{10}$$

Taking expectations and using $\mathbb{E}[(x - \mu)] = 0$ and $\mathbb{E}[(x - \mu)^2] = \sigma^2$ leads to:

$$\mathbb{E}[f(x)] = f(\mu) + \frac{1}{2}f''(\mu)\sigma^2 + \cdots \tag{11}$$

$$\mathbb{E}[f(x)^2] = f(\mu)^2 + f'(\mu)^2\sigma^2 + f(\mu)f''(\mu)\sigma^2 + \cdots \tag{12}$$

Substituting these into the definition of the variance, and simplifying while retaining only the leading-order term, we obtain:

$$\mathrm{Var}[f(x)] = (f(\mu)^2 + f'(\mu)^2\sigma^2 + f(\mu)f''(\mu)\sigma^2 + \cdots) - (f(\mu) + \frac{1}{2}f''(\mu)\sigma^2 + \cdots)^2 \tag{13}$$

$$= (f(\mu)^2 + f'(\mu)^2\sigma^2 + f(\mu)f''(\mu)\sigma^2 + \cdots) - (f(\mu)^2 + f(\mu)f''(\mu)\sigma^2 + \cdots) \tag{14}$$

$$\approx f'(\mu)^2\sigma^2 \tag{15}$$

which completes the proof.

Finally, a Taylor expansion of the Sigmoid and Gompertz gate functions for large positive input values demonstrates that these two functions converge to each other exponentially fast in this regime, as pointed out in Section 2.1.

$$\mathrm{Sigmoid}(x) - \mathrm{Gompertz}(x) = \frac{1}{1 + e^{-x}} - e^{-e^{-x}} \tag{16}$$

$$= \left(1 - e^{-x} + \mathcal{O}(e^{-2x})\right) - \left(1 - e^{-x} + \mathcal{O}(e^{-2x})\right) \tag{17}$$

$$= \mathcal{O}(e^{-2x}) \tag{18}$$

## B   Flipped Mish: a new self-gated activation with right-leaning distribution

Throughout this work, we have emphasized the influence of right-skewed asymmetry in the underlying distribution associated with an activation on model performance. To further explore this property, we leverage the left-skewed distribution underlying Mish to construct a new self-gated activation function exhibiting right-skewed asymmetry. This is achieved by reflecting the Mish-associated distribution about the vertical axis. Specifically, denoting the Mish distribution by $D(x)$ and its corresponding gate function by $g(x)$, the gate function for the flipped distribution $\tilde{D}(x) = D(-x)$ is given by $\tilde{g}(x) = 1 - g(-x)$, as shown by the following derivation:

$$\tilde{g}(x) = \int_{-\infty}^{x} dy\, \tilde{D}(y) = \int_{-\infty}^{x} dy\, D(-y) = \int_{-x}^{\infty} dy\, D(y) = \int_{-\infty}^{\infty} dy\, D(y) - \int_{-\infty}^{-x} dy\, D(y) = 1 - g(-x) \tag{19}$$

where in the third equation we have redefined the dummy integration variable $y \to -y$ and in the last equation we have used the fact that the integral of the distribution $D(y)$ over its entire domain is equal to $1$. The resulting activation function, which we refer to as Flipped Mish (FMish), is thus defined as:

$$\mathrm{FMish}(x) = x(1 - \tanh(\mathrm{softplus}(-x))) \tag{20}$$

Figure 8 compares FMish with GoLU, including their respective gate functions and the associated distributions.

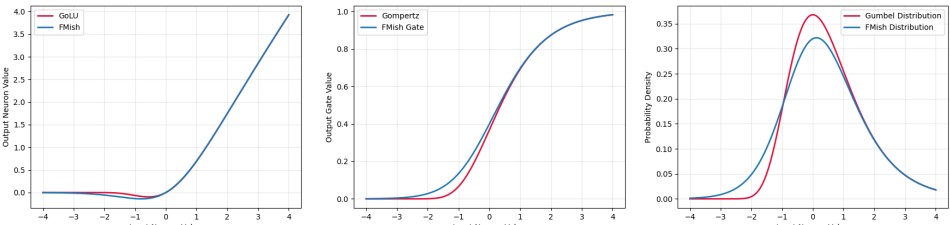

Figure 8: Comparison of activations (left), gate functions (middle), and associated distributions (right) for FMish and GoLU.

We evaluated FMish on ResNet18, ResNet50 and ViT-B/32 trained on ImageNet-1k. Remarkably, it outperformed all baseline activations except GoLU, achieving $70.73\pm0.05$ for ResNet18, $76.20\pm0.01$ for ResNet50 and $75.67\pm0.04$ for ViT-B/32 (compare with results in Table 2 ). This outcome aligns with expectations, as the slope of FMish at the origin is 0.4, lower than that of sigmoid and Gaussian CDF (0.5) but slightly higher than GoLU (0.37). These results further highlight the significance of right-leaning asymmetry and the resulting variance reduction.

Furthermore, we note that the Flipped Mish distribution does not decay as rapidly as the Gumbel distribution for large negative inputs, which may also contribute to its performance.

## C Details of the loss landscape experiment

We analyze the loss landscape of a neural network by quantitatively measuring and visualizing how the loss changes as the network's parameters are perturbed. Smoothness in the loss landscape often indicates that small perturbations in the parameters do not cause large changes in the loss, which can make optimization more stable.

Specifically, we generate two random perturbation directions $d_1$ and $d_2$, each matching the shape of the model parameters. The elements of these directions are independently sampled from a Standard Normal distribution. To ensure controlled magnitudes, each perturbation direction is subsequently normalized.

We perturb the weights of the model along these directions in a linear combination:

$$W_{\text{perturbed}} = W_{\text{trained}} + \alpha d_1 + \beta d_2 \tag{21}$$

where $W_{\text{trained}}$ are the trained weights of the model and $\alpha$ and $\beta$ are scalar values that determine the perturbation magnitude and are chosen as $\alpha, \beta \in [-1, 1]$. For each pair of values $(\alpha, \beta)$, we compute the loss using the perturbed weights $W_{\text{perturbed}}$ on the full test set of the CIFAR-10 dataset. We then repeat this for a grid of $(\alpha, \beta)$ values to create a 3D surface plot as shown in Figure 5.

To provide a more quantitative understanding of the loss landscapes in Figure 5, we have plotted the density functions of the loss values for each activation function and computed their variance. The results, shown in Figure 9, indicate that GoLU achieves both a lower average loss and smaller variance compared to other activations, consistent with the observations from the 3D plots in Figure 5.

## D Learning Rate ablation

For various tasks, we conduct a focused search over the learning rate to determine whether the default setting represents the optimal value and to assess its impact on the performance of models trained with different activation functions. Figures 10 and 11 present heatmaps of test results for Semantic Segmentation and Diffusion tasks, comparing models trained with various activation functions across different learning rates. Figures 12 and 13 show similar heatmaps for the Instance Segmentation task, reporting box mAP and mask mAP, respectively. For these tasks, the default learning rate, highlighted by a black box, differs from the optimal learning rate, indicated by a green box. Notably, while GoLU performs slightly below the best-performing activation under the default learning rate, it outperforms all other activation functions when evaluated at the optimal learning rate, which is consistent across all activations.

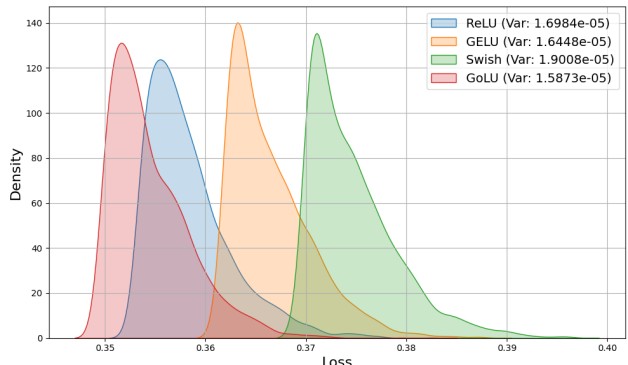

Figure 9: Comparison of loss value distributions across the loss landscape. GoLU achieves both a lower loss mean and variance.

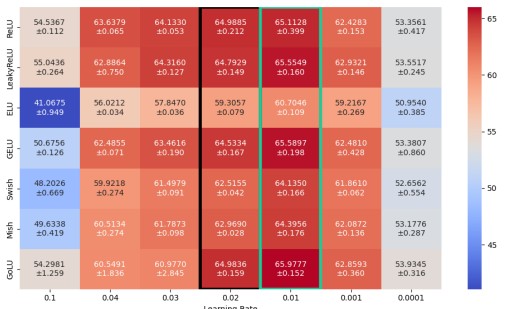

Figure 10: Test mIoU - DeepLabV3 on MS-COCO. The default learning rate is 0.02 which is colored in black and the best learning rate is 0.01 which is colored in green.

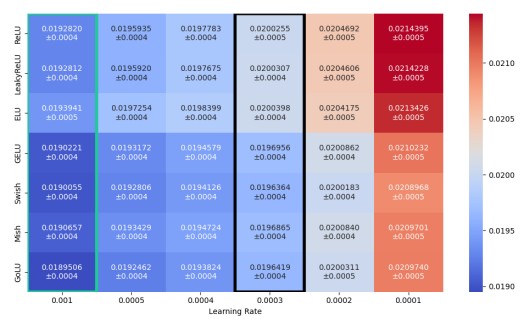

Figure 11: Test Loss - DDPM on CelebA. The default learning rate is 0.0003 which is colored in black and the best learning rate is 0.001 which is colored in green.

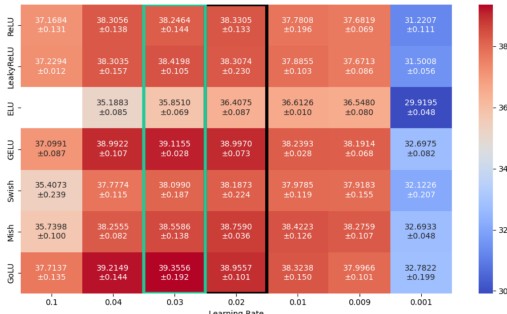

Figure 12: Test Box mAP - Mask R-CNN-PFN ResNet-50 on MS-COCO. The default learning rate is 0.02 which is colored in black and the best learning rate is 0.03 which is colored in green.

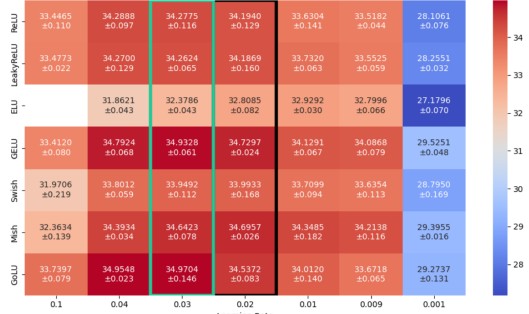

Figure 13: Test Mask mAP - Mask R-CNN-PFN ResNet-50 on MS-COCO. The default learning rate is 0.02 which is colored in black and the best learning rate is 0.03 which is colored in green.

Motivated by these results, we further investigate the impact of learning rate on image classification tasks where GoLU demonstrated superior performance compared to baseline activations. Figures 14 and 15 present heatmaps of test accuracies for ResNet-50 and ViT-B/32 on ImageNet-1k.

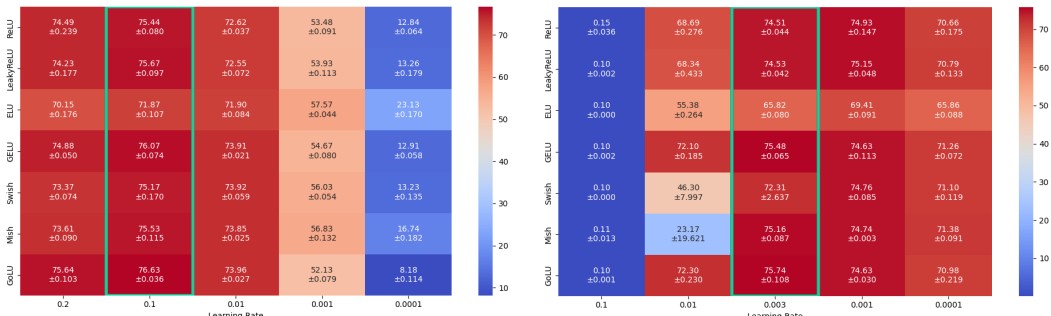

Figure 14: Test accuracies - ResNet-50 on ImageNet-1k. The default learning rate is 0.1 which is also the best and is colored in green.

Figure 15: Test accuracies - ViT-B/32 on ImageNet-1k. The default learning rate is 0.003 which is also the best and is colored in green.

Notably, we observe that the optimal learning rate aligns with the default learning rate in this case. These findings reinforce the broader trend that, with few exceptions, *GoLU consistently outperforms baseline activation functions across tasks when evaluated at the optimal learning rate*.

# E   Critical Difference Analysis

In this section, we conduct a Critical Difference analysis following Demšar [2006] to systematically rank activation functions based on experiments performed on ImageNet-1k, MS-COCO, OWT, TS, and CelebA. As shown in Figure 16, GoLU achieves the highest rank, followed by GELU. Notice that the confidence interval in this analysis is independent of the variance across multiple runs with different random seeds. Instead, it is determined by the number of models and datasets, as well as the significance level, which is set to $\alpha = 0.05$ here.

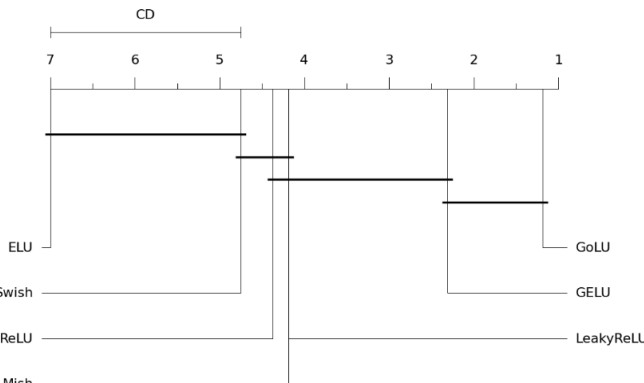

Figure 16: Critical Difference diagram, ranking activation functions based on average performance.

# F   Training and inference times

Table 9 reports relative training and inference times with respect to baseline activations for our trained models. On average, GoLU achieves training and inference speeds on par with default activation functions, while offering improved performance. This makes GoLU a practical and effective alternative for training deep learning models.

# G   Experimental Details

This section outlines detailed information about the datasets and training pipelines used for the various tasks studied in this work. All experiments in this section were conducted on NVIDIA A100

Table 9: Relative training and inference time with respect to baseline activations for our trained architectures.

| Architecture | Dataset | Baseline Activation | Relative Training Speed | Relative Inference Speed |
|---|---|---|---|---|
| ResNet-18 | ImageNet-1k | ReLU | 1.00x | 1.00x |
| ResNet-34 | ImageNet-1k | ReLU | 1.01x | 1.00x |
| ResNet-50 | ImageNet-1k | ReLU | 1.01x | 1.01x |
| WideResNet-50-2 | ImageNet-1k | ReLU | 1.03x | 1.02x |
| DenseNet-121 | ImageNet-1k | ReLU | 1.02x | 1.02x |
| EfficientNet-B0 | ImageNet-1k | Swish | 1.00x | 1.00x |
| TinyViT | ImageNet-1k | GELU | 0.99x | 0.98x |
| ViT-B/32 | ImageNet-1k | GELU | 0.99x | 0.99x |
| ViT-B/16 | ImageNet-1k | GELU | 0.98x | 0.98x |
| babyGPT | TinyStories | GELU | 1.00x | 1.00x |
| GPT2-S | OpenWebText | GELU | 1.01x | 1.01x |
| DeepLabV3 | MS-COCO | ReLU | 1.14x | 1.04x |
| RetinaNet | MS-COCO | ReLU | 1.00x | 1.00x |
| FasterRCNN | MS-COCO | ReLU | 1.03x | 1.00x |
| MaskRCNN | MS-COCO | ReLU | 1.05x | 1.02x |
| DDPM | CelebA | Swish | 0.97x | 0.97x |
| Average | - | - | 1.01x | 1.00x |

GPUs, with an approximate total compute time of 112K GPU hours, except for TinyViT, which was executed on an NVIDIA H100 GPU with a total runtime of 455 GPU hours.

## G.1  Image Classification - ImageNet

In image classification experiments on ImageNet-1k, ResNets 18, 34, 50, WideResNet-50-2 and DenseNet-121 are trained for 90 epochs with a batch size of 256, SGD with momentum=0.9 (Nesterov for WRN-50-2 and DN-121), learning rate 0.1, and weight decay $1 \times 10^{-4}$. Further, a Step learning rate scheduler is applied that reduces the learning rate by a gamma = 0.1 after every 30 epochs. EfficientNet-B0 is trained using the timm library for 450 epochs with a batch size of 1536 using RMSProp Hinton et al. [2012] with an initial learning rate of 0.048 and a weight decay of $1 \times 10^{-5}$. ViT models are trained for 300 epochs with a batch size of 4096 using AdamW Loshchilov and Hutter [2017] with an initial learning rate of $3 \times 10^{-3}$ and weight decay of 0.3. Various regularization techniques are applied, including Exponentially Moving Averaged Weights Tarvainen and Valpola [2017], AutoAugment Cubuk et al. [2019] (ImageNet policy for ViTs), RandAugment Cubuk et al. [2020], MixUp Zhang et al. [2017], CutMix Yun et al. [2019] and Label Smoothing Szegedy et al. [2016] for EfficientNet-B0 and ViT models. ViT-B/16 shows slight instability for seed 1 for GELU. Hence we further average seeds 2 and 3 for both GELU and GoLU. We find that GELU shows a top-1 accuracy of $80.61 \pm 0.06$ while GoLU shows top-1 accuracy of $80.69 \pm 0.07$ which is higher than GELU.

## G.2  Image Classification - CIFAR-10

The ResNet 20, 32, 44, 56 and 110 models are trained for 164 epochs with a batch size of 128, a learning rate of 0.1, and SGD with momentum 0.9. A weight decay of $1 \times 10^{-4}$ is applied, along with a MultiStep learning rate scheduler with a gamma factor of 0.1 at epochs 81 and 122 (with an initial learning rate of 0.01 and additional gamma factor of 10 at epoch 2 for ResNet-110).

WideResNet28-2 and DenseNet40, were trained for 200 and 300 epochs, and batch sizes of 128 and 64, respectively. We employ SGD with Nesterov momentum 0.9 for both architectures, using a learning rate of 0.1. The weight decays are $5 \times 10^{-4}$ for WideResNet28-2 and $1 \times 10^{-4}$ for DenseNet40. Similar to ResNets, both WideResNet28-2 and DenseNet40 use the MultiStep learning rate scheduler. However, WideResNet28-2 reduces the learning rate by a factor of 0.2 at epochs 60, 120, and 160, while DenseNet40 reduces the learning rate by 0.1 at epochs 150 and 225. To train ViT-Ti/16-224 from scratch, we leverage the Timm library.

### G.3 Language modeling

Both, TinyStories and OpenWebText datasets are popular benchmarks for training language models. The TinyStories dataset consists of 2,119,719 data points in the training set and 21,990 in the test set, while the OpenWebText dataset has 8,009,762 data points in the training set and 4,007 data points in the test set. Both babyGPT and nanoGPT have a vocabulary size of 50,304 and a maximum sequence length of 1024.

The babyGPT version of the GPT-2 series consists of 6 layers, 6 attention heads, and an embedding dimension of 384, with a feed-forward expansion dimension of 1536 output features. The model is trained for 10,000 iterations with a batch size of 640, using the AdamW optimizer. The initial learning rate is $1 \times 10^{-3}$, with a minimum learning rate of $1 \times 10^{-4}$, a weight decay of 0.1, and a gradient clipping norm of 1.0. A Cosine learning rate scheduler is applied with a linear warmup for the first 100 iterations.

Similarly, the GPT2-S model consists of 12 layers, 12 attention heads, and an embedding dimension of 768. It trains for 600,000 iterations with a batch size of 480, using the AdamW optimizer (with $\beta_2 = 0.95$). The initial learning rate is $6 \times 10^{-4}$, with a minimum learning rate of $6 \times 10^{-5}$, a weight decay of 0.1, and a gradient clipping norm of 1.0. The Cosine learning rate scheduler is employed with a linear warmup for the first 2,000 iterations.

### G.4 Semantic Segmentation

The MS-COCO dataset with PASCAL-VOC labels contains 92,518 data points in the training set and 5,000 data points in the test set. The original MS-COCO dataset contains 117,266 data points in the training set. However, the existing benchmark pre-processes and removes images that either lack valid annotations or contain only small objects with an area coverage of less than 1,000 pixels. This ensures the retention of meaningful data points for training the model.

The DeepLabV3-ResNet-50 model is trained for 30 epochs with a batch size of 32, using SGD with momentum 0.9, a learning rate of $2 \times 10^{-2}$, weight decay of $1 \times 10^{-4}$, and a polynomial learning rate scheduler with a power of 0.9.

### G.5 Object Detection

Unlike Semantic Segmentation, the MS-COCO dataset for object detection contains 117,266 images in the training set and 5,000 images in the test set. Additionally, we do not apply any pre-processing that removes images from the training or test sets.

Faster R-CNN-FPN ResNet-50 and RetinaNet-FPN ResNet-50 are trained for 26 epochs with a batch size of 16, an aspect ratio group factor of 3, no frozen batch normalization, and a MultiStep learning rate scheduler that reduces the initial learning rate by a factor of 0.1 at epochs 16 and 22. Specifically, Faster R-CNN-FPN ResNet-50 uses SGD with momentum 0.9, a learning rate of $2 \times 10^{-2}$, and a weight decay of $1 \times 10^{-4}$, while RetinaNet-FPN ResNet-50 uses the AdamW optimizer with a learning rate of $1 \times 10^{-4}$ and a weight decay of $5 \times 10^{-2}$.

### G.6 Instance Segmentation

The MS-COCO dataset for instance segmentation uses the same train and test sets as those used for Object Detection. Additionally, it trains with the exact same configurations used for Faster R-CNN-FPN in the previous subsection G.5.

### G.7 Denoising Diffusion Probabilistic Models

The CelebA dataset, comprises of 162,770 training images and 19,867 test images of human faces. The Denoising Diffusion Probabilistic Model is trained on the CelebA dataset for 50 epochs with a batch size of 32 leveraging the DDPM Kim [2023] repository. The AdamW optimizer with a learning rate of 0.0003, Cosine learning rate scheduler, and linear learning rate warmup for the first 1,000 iterations are applied.

# H Test Loss Curves

To provide a more comprehensive view of GoLU's test performance over the course of training, this section presents test curves, including loss and task-specific metrics, comparing GoLU with ReLU and GELU, and illustrating how their performance evolves throughout training.

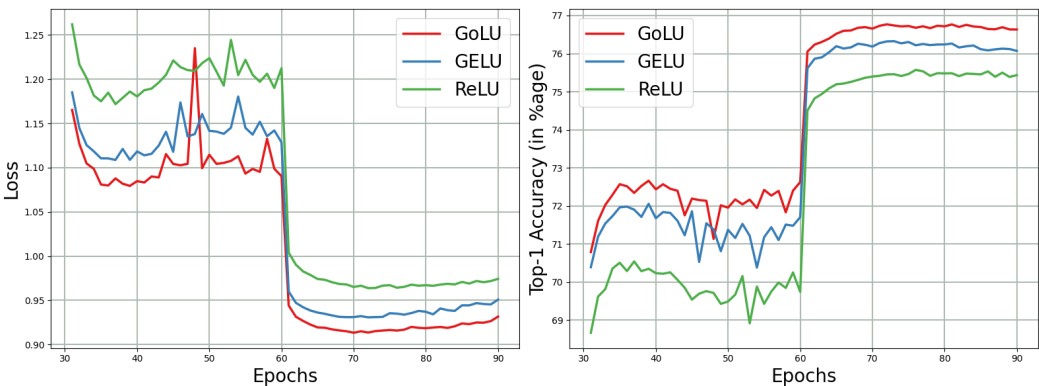

Figure 17: ResNet-50 test loss (Left) and test top-1 accuracy (Right) on ImageNet-1k.

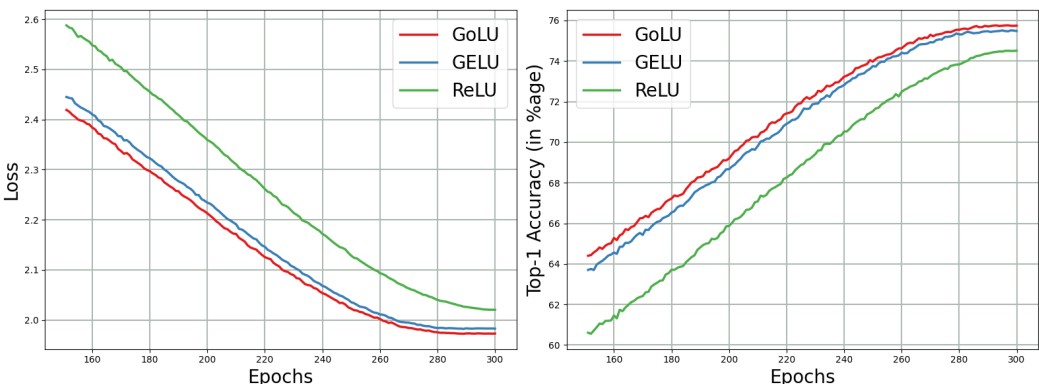

Figure 18: ViT-B/32 test loss (Left) and test top-1 accuracy (Right) on ImageNet-1k.

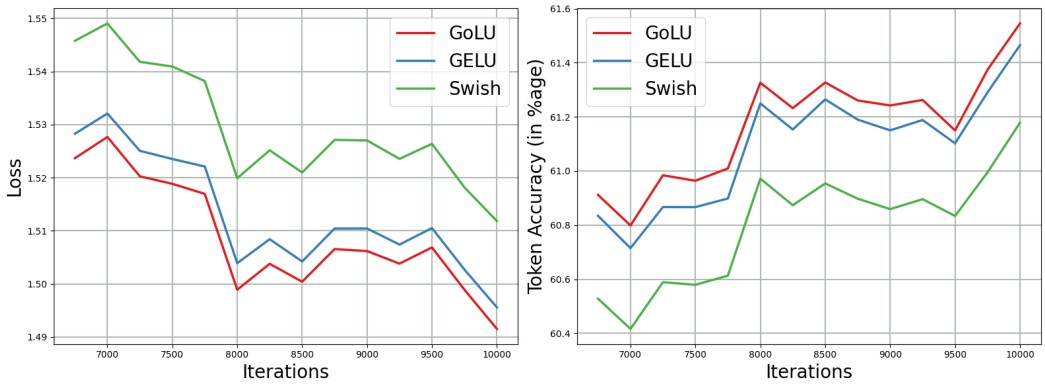

Figure 19: babyGPT test loss (Left) and test token accuracy (Right) on TS.

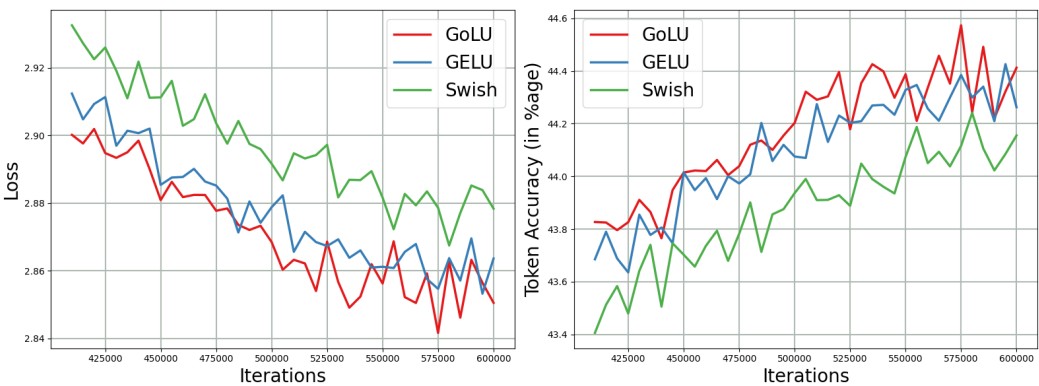

Figure 20: GPT2-S test loss (Left) and test token accuracy (Right) on OWT.

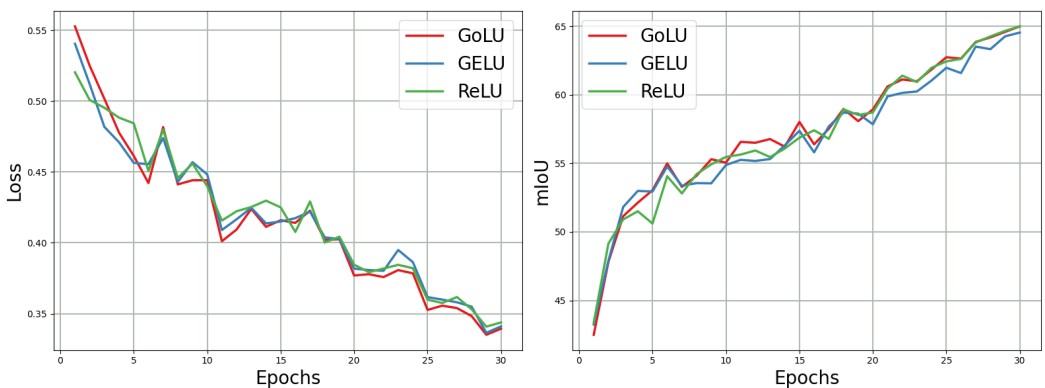

Figure 21: DeepLabV3 ResNet-50 test loss (Left) and test mIoU (Right) on MS-COCO with lr=0.02.

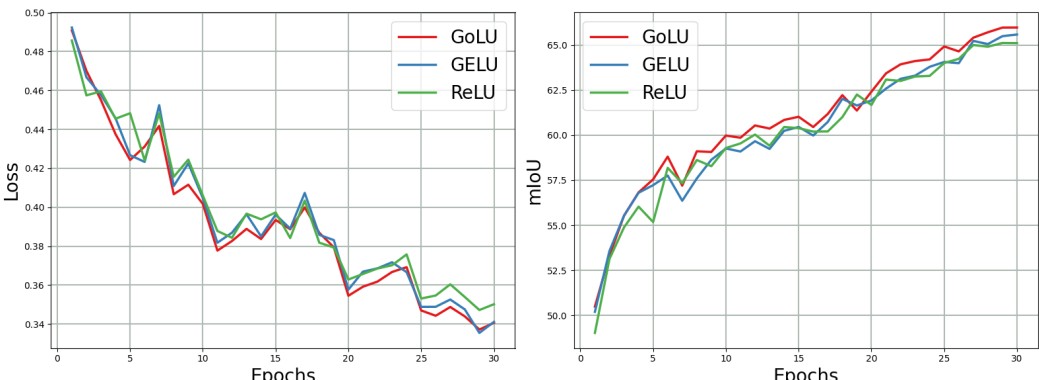

Figure 22: DeepLabV3 ResNet-50 test loss (Left) and test mIoU (Right) on MS-COCO with lr=0.01.

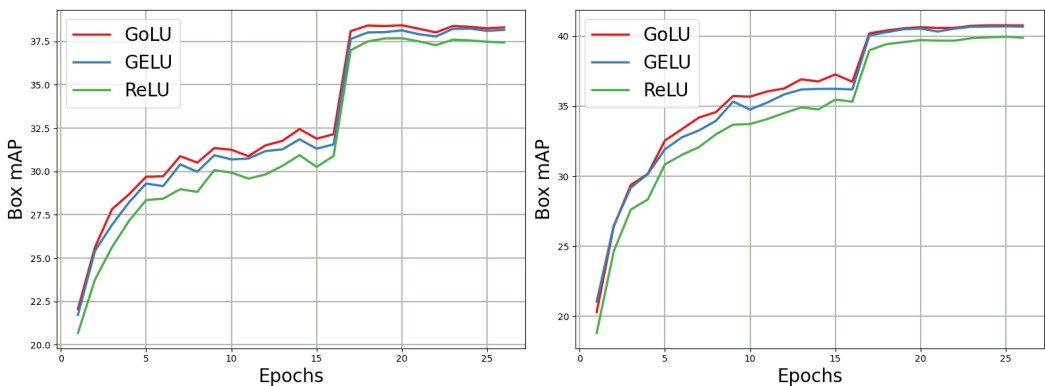

Figure 23: Faster R-CNN-FPN ResNet-50 (Left) and RetinaNet-FPN ResNet-50 (Right) test Box mAP on MS-COCO.

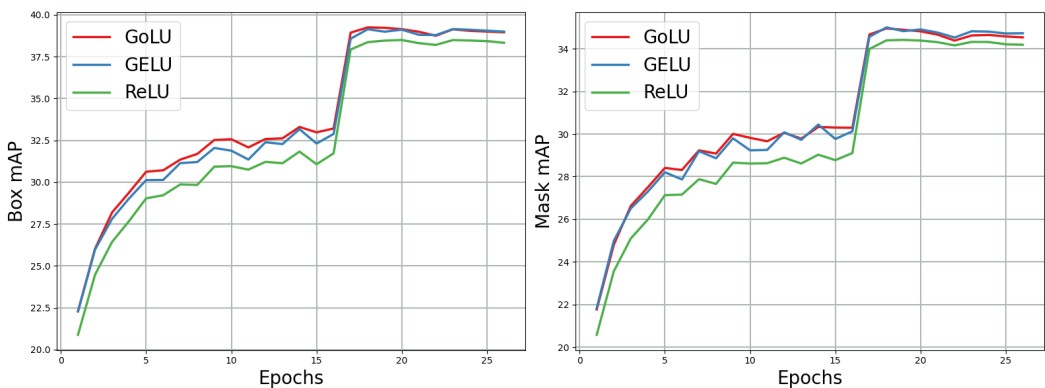

Figure 24: Test Box mAP (Left) and test Mask mAP (Right) for Mask R-CNN-FPN ResNet-50 trained on MS-COCO.

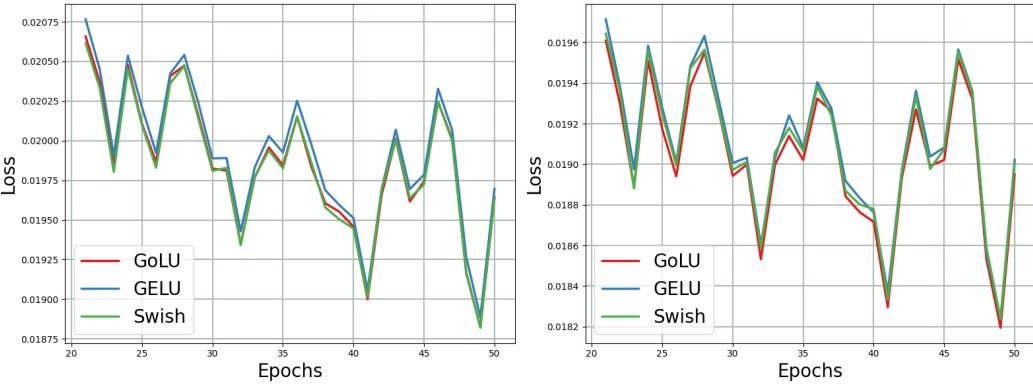

Figure 25: Test loss for Denoising Diffusion Probabilistic Model trained on CelebA at LR=0.0003 (Left) and LR=0.001 (Right).

# I   Machine Translation

To further assess GoLU across diverse tasks, we evaluated its performance on machine translation using the WMT14 English–German benchmark, with approximately 4.5 million training pairs. Specifically, we trained Transformer-Big models using the Fairseq framework [Ott et al., 2019], comparing GoLU against baseline activations including ReLU, which is the default in this architecture. The architecture follows the standard configuration with 6 encoder and 6 decoder layers, 1024-dimensional embeddings, 16 attention heads, and a feed-forward hidden size of 4096. Models

were trained with three different random seeds for 50 epochs using the Adam optimizer ($\beta_1 = 0.9$, $\beta_1 = 0.98$), an inverse square root learning rate schedule (base LR $= 5 \times 10^{-4}$, 4000 warm-up steps), label smoothing of 0.1, and gradient accumulation of 16 to simulate large-batch training. Evaluation was conducted using beam search with BLEU4 as the performance metric. All runs were executed on a single NVIDIA L40S GPU with a total runtime of roughly 1750 GPU hours. As shown in Table 10, GoLU outperforms standard activation functions in terms of mean BLEU4 score, which highlights its effectiveness in sequence modeling as well.

Table 10: Mean and standard error of BLEU4 scores for Transformer-Big on the WMT14 English–German translation task.

| Activation | Test BLEU4 |
|---|---|
| ReLU | 28.33±0.14 |
| LeakyReLU | 28.26±0.04 |
| ELU | 27.49±0.08 |
| GELU | 28.20±0.04 |
| Swish | 28.34±0.08 |
| Mish | 28.31±0.10 |
| GoLU | **28.44±0.15** |

## J   Case Study: Bayesian Learning Curve Extrapolation using Prior-data fitted Networks

In this section, we present an additional experiment on GoLU, initially conducted as an internal validation study. We report this as a "negative" result, with GoLU ranking second-to-last under the optimal learning rate. Due to the unconventional experimental setup, its niche focus, and suboptimal hyperparameter tuning, we have included these findings in the appendix rather than in the main text.

**Experimental Details**   In this experiment, we assessed all 7 activation functions (including GoLU) considered in the main article as activations for LC-PFN Adriaensen et al. [2024]. LC-PFN is a prior-data fitted network Müller et al. [2021] that functions as a decoder-only transformer, trained for in-context Bayesian prediction for a specific prior dataset distribution. Specifically, LC-PFN is trained for Bayesian Learning Curve extrapolation. We adopted the same setup used to train the best model presented in the original paper, a decoder-only transformer having 26.79M trainable parameters, 12 layers, 4 attention heads, an embedding dimension of 512, and a feed-forward expansion dimension of 1024 output features. It was trained using 10M synthetically generated learning curves, (each containing 100 observations), employing the Adam optimizer (with a default learning rate of 0.0001 and a batch size of 100), using a cosine scheduler with a linear warmup during the first 25,000 steps (25%) of the training. At test time, it takes a partial learning curve as input, and predicts the posterior predictive distribution (PPD) for possible continuations. The test performance of the final model was measured using the log-score, which represents the log-likelihood of the true continuation, under the PPD, averaged across all 99 cutoffs for 10,000 curves from the prior.

**Results**   Figure 26 presents the log-scores for the final models, utilizing all 7 activation functions at 5 different learning rates, averaged over 3 training runs. At the original and optimal learning rate of 0.0001, GoLU ranks 6th among the 7 activations. However, a closer examination reveals that the choice of activation function seems to have minimal impact, as the differences between GoLU and the best (ELU) and worst (Swish) activation are within a single standard error. The learning rate ablation shows that GoLU ranks first at the highest stable learning rate (0.001), supporting previous findings that GoLU thrives in the high learning rate regime.

## K   Limitations

The effects attributed to GoLU, as described in Section 2.2, are not guaranteed to hold universally but rather represent general trends observed in our empirical findings. Moreover, while asymmetry has been highlighted as a distinctive feature of GoLU, its high performance, detailed in Section 3,

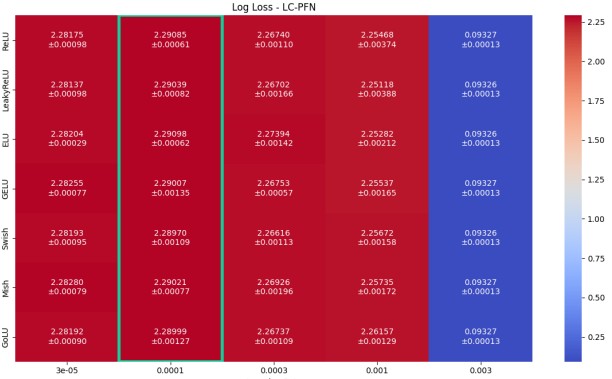

Figure 26: Test log scores - LC-PFN. The default learning rate is 0.0001, which is also optimal, is highlighted in green.

cannot be solely attributed to asymmetry, but instead arises from an intricate interplay of properties described in Section 2.1.

Additionally, we note the importance of learning rate tuning in realizing the full benefits of GoLU. Although we generally observed that, under a tuned learning rate, which also improves the performance of all baseline activations, GoLU achieves superior results, it is not guaranteed to outperform baseline activation functions under default hyperparameter settings.

## L   Broader Impact

This paper introduces GoLU, a novel activation function designed to advance the field of Machine Learning. The primary objective of this work is to improve the performance and robustness of state-of-the-art neural networks across diverse domains, including computer vision, natural language processing, and generative modeling. The societal impact of this work is primarily tied to the downstream applications of machine learning models that may incorporate GoLU. By enhancing the robustness and performance of models, our activation function has the potential to positively influence critical areas such as medical imaging, autonomous systems, and other technologies that drive societal progress. While there are no immediate or direct societal concerns specific to GoLU itself, as with any development in machine learning, there is a possibility of misuse. We therefore emphasize the importance of ethical and responsible deployment of machine learning technologies enhanced by our contributions.

