# OpenReview forum: "Gompertz Linear Units: Leveraging Asymmetry for Enhanced Learning Dynamics"
_NeurIPS.cc/2025/Conference — NeurIPS 2025 poster_

### Official Review · Reviewer_AMoC · 2025-06-30

**Clarity:** 3
**Significance:** 3
**Originality:** 3
**Rating:** 5
**Confidence:** 3

**Summary:**

The paper proposes the Gompertz Linear Unit (GoLU), a smooth, self-gated activation function defined as GoLU(x) = x · e^{-e^{-x}}, where the gate is the Gompertz CDF of a standard Gumbel distribution. This form introduces a right-skewed asymmetry that reduces the slope near zero, encouraging lower activation variance while ensuring stable gradient flow. The authors analytically link this behavior to implicit regularization and empirically show that GoLU produces tighter latent representations and more expressive weights, resulting in smoother loss landscapes. Across a wide range of vision, language, and translation benchmarks, GoLU consistently outperforms existing activations like ReLU, GELU, Swish, and Mish, all while maintaining efficiency through a custom CUDA kernel.

**Questions:**

- **Asymmetry Ablation** – Could you run an ablation to isolate the role of GoLU’s asymmetry? For example, (a) evaluate a symmetric version such as ½ [GoLU(x) + GoLU(−x)], and (b) rescale GoLU to match GELU’s slope near zero. This would help clarify whether the asymmetry is essential for the observed gains.
- **Interaction with Optimizers** – Since modern optimizers like Adam and Lion are designed to stabilize gradients, how does GoLU interact with them? It seems that the current experiments are conducted with momentum-based optimizers. I’m wondering whether GoLU’s effect—such as producing a smoother loss landscape—would become more prominent with vanilla SGD or be overshadowed by specific optimizer dynamics.

**Ethical Concerns:**

["NO or VERY MINOR ethics concerns only"]

**Final Justification:**

The authors' rebuttal successfully addressed my concerns. I have raised my score from 4 to 5.

**Limitations:**

Yes

**Paper Formatting Concerns:**

I have reviewed the paper and did not find any significant formatting issues.

**Quality:**

3

**Strengths And Weaknesses:**

**Strengths**

- The proposed Gompertz Linear Unit (GoLU) introduces a novel activation function with a self-gating mechanism based on the Gompertz CDF, offering a fresh addition to the family of smooth activations.
- The paper provides solid theoretical support, showing that GoLU’s reduced slope near zero leads to implicit regularization and smoother loss landscapes. The analytical derivations are thorough and well-motivated.
- The authors conduct extensive empirical validation across diverse domains—including vision, language modeling, and machine translation—demonstrating consistent improvements over standard activations such as ReLU, GELU, Swish, and Mish.
- The paper is clearly written and well-structured.

**Weaknesses**

- While performance improvements are consistent across tasks, they are generally small in magnitude. This limited headroom may reduce the practical impact and adoption of GoLU, especially given the added complexity of a new activation function.

---

> ### Author Rebuttal · Authors · 2025-07-30
>
> We sincerely thank the reviewer for their constructive feedback and questions. Below, we address the points raised in the review.
>
> **Weakness:**
> While we understand the reviewer’s point that performance improvements may appear small in some cases, we respectfully argue that even fractional improvements in accuracy can lead to substantial impact especially when amortized across many different deep learning applications. Activation functions like GELU and Swish also showed small improvements over ReLU, but they are now widely adopted, improving results in thousands of code bases. We specifically emphasize that, unlike many activations that show task or architecture-specific benefits, GoLU provides more consistent improvements across domains, including vision, language modeling, and generative modeling, as demonstrated in our experiments.
>
> We would also like to emphasize a trend we observed in our Table 2 that the difference in top-1 accuracy between GoLU and GELU on ImageNet increases for larger models:
>
> - RN18: +0.10
> - RN34: +0.27
> - RN50: +0.56
> - WRN50-2: +0.65
>
> In addition to its empirical performance, GoLU has a simple, closed-form definition and is theoretically well-motivated. Our provided CUDA kernel enables easy integration with no added latency, making GoLU a practical activation function that can be widely adopted by the community.
>
> **Question on Asymmetry Ablation:**
> We appreciate the reviewer’s suggestion to ablate the role of asymmetry. The specific example, ½ [GoLU(x) + GoLU(−x)] will symmetrize the full activation which is not desirable, however one can symmetrize the underlying distribution which will lead to ½ x [1+Gompertz(x) -Gompertz(−x)] (the derivation follows similar lines to our eq.19). This does not enjoy the properties of GoLU and is expected to lead to lower performance.
>
> Instead, in Appendix B, we have followed a relatively similar idea, but have taken advantage of the insights gained in this work to explore whether we can *improve* an existing activation. Specifically, we took Mish, which is based on a left-leaning distribution, and flipped its distribution to obtain an activation with right-leaning asymmetry, which we refer to as Flipped Mish (FMish). We observed that this significantly improves performance compared to the original Mish activation and, surprisingly, makes FMish outperform symmetric activations like Swish and GELU, while it still does not surpass GoLU. This result aligns closely with what their slopes at the origin suggest: the slope of FMish (0.4) is lower than Mish (0.6) and Swish/GELU (both 0.5), but still slightly higher than that of GoLU (0.37).
>
> Additionally, following the reviewer’s suggestion, we also conducted a more comprehensive ablation on the slope of GoLU. We introduced a parameter $\beta$ which controls the slope $\mathrm{GoLU}_{\beta}(x) = x e^{-\beta e^{-x}}$ and conducted an ablation study on this parameter for ResNet-34, ResNet-50, ViT-B/32, and ViT-B/16. The results, provided in the table below, indicate that for all these models, decreasing $\beta$ below 1 (which increases the slope) degrades performance, aligning with our expectations. Moreover, for ResNet-34, increasing the $\beta$ parameter to 1.5 or 2 (which decreases the slope) improves performance before degrading again at higher values. This shows that the default value ($\beta=1$) provides a balanced variance control in most cases, although some architectures may benefit from further reducing the slope.
>
> | β     | ResNet34            | ResNet50            | ViT-B/32           | ViT-B/16           |
> |-------|---------------------|---------------------|--------------------|--------------------|
> | 0.10  | 69.86 ± 0.044       | 71.26 ± 0.167       | 70.87 ± 0.116      | 77.34 ± 0.014      |
> | 0.50  | 73.18 ± 0.030       | 75.84 ± 0.086       | 75.52 ± 0.150      | 80.41 ± 0.048      |
> | 1.00  | 73.71 ± 0.043       | **76.63 ± 0.036**       |**75.74 ± 0.108**      |**80.72 ± 0.052**      |
> | 1.50  | **73.86 ± 0.050**   | 76.50 ± 0.148       | 75.70 ± 0.018      | 80.40 ± 0.056      |
> | 2.00  | **73.86 ± 0.074**   | 76.43 ± 0.147       | 75.58 ± 0.011      | 80.33 ± 0.051      |
> | 5.00  | 72.92 ± 0.056       | 73.03 ± 0.153       | 75.21 ± 0.070      | 80.00 ± 0.086      |
> | 10.00 | 70.26 ± 0.173       | 66.36 ± 0.142       | 74.54 ± 0.088      | 79.56 ± 0.031      |
>
> **Question on Interaction with Optimizers:**
> While we did use momentum-based optimizers in many of our vision experiments, we also evaluated GoLU with adaptive optimizers such as AdamW in multiple settings:
> - TinyViT, ViT-B/32 and ViT-B/16 on ImageNet-1k
> - GPT language models on TinyStories and OWT
> - Denoising Diffusion Probabilistic Model on CelebA
>
> In all of these cases, GoLU outperformed baseline activations, which suggest that GoLU’s improved generalization complements adaptive optimizers as well.
>
> Additionally, as an example of a model trained with AdamW, we selected babyGPT trained on TinyStories and conducted the same loss landscape analysis as in Figures 5 and 9. We observed the same qualitative behavior, with GoLU exhibiting both the lowest mean loss and the lowest variance across the landscape:
>
> | Activation | Mean Loss     | Loss Variance     |
> |------------|-----------|--------------|
> | Swish      | 1.6301    | 3.94e-05     |
> | ReLU       | 1.6259    | 3.93e-05     |
> | GELU       | 1.6133    | 3.93e-05     |
> | GoLU       | **1.6079**    | **3.91e-05**     |
>
> We hope to have addressed the concerns raised by the reviewer. We respectfully ask that if you feel more positively about our paper, you kindly reconsider your rating accordingly. If not, please let us know if you have further questions or what can be further improved. We are happy to continue the discussion at any time before the end of the discussion period on August 6th. Thank you.

---

> > ### Comment · Reviewer_AMoC · 2025-08-05
> >
> > Thank you for the helpful clarifications. I will revise my score in light of them.

---

> > > ### Author Response · Authors · 2025-08-08
> > >
> > > We sincerely thank the reviewer for their constructive feedback and suggestions. We greatly appreciate their decision to reassess the paper and raise their score.

---

### Official Review · Reviewer_8T3r · 2025-07-02

**Clarity:** 3
**Significance:** 3
**Originality:** 3
**Rating:** 5
**Confidence:** 3

**Summary:**

The document introduces the Gompertz Linear Unit (GoLU), a novel activation function for deep learning that leverages the Gompertz function as its gating mechanism. Its asymmetry, derived from the Gumbel distribution, enables reduced activation variance and preserves robust gradient flow. The experiments show that this activation function can help enhance model robustness and generalization.

**Questions:**

I don't have question towards the current shape of the paper but I hope the authors can provide more insights on future development.

**Ethical Concerns:**

["NO or VERY MINOR ethics concerns only"]

**Limitations:**

yes.

**Quality:**

3

**Strengths And Weaknesses:**

Quality: The paper conducts extensive experiments on a lot of tasks, including image classification, language modeling, video segmentation and even image generation (diffusion models), showing substantial efforts in demonstrating the effectiveness of the proposed algorithm. The proposed activation function seems to be able to outperform existing activation functions by a consistent margin across different tasks.

Clarity: The writing of this paper is clear (partially due to the simplicity of this paper), easy to follow with a substantial level of details. It would be quite straight-forward to replicate the authors' experiments.

Significance: the experiment results look very promising as the advantages are clear and the margin are significant and consistent. It would be great if the authors can provide more insights, e.g. is there any good principle that can help further improve performance?

---

> ### Author Rebuttal · Authors · 2025-07-30
>
> We sincerely thank the reviewer for their positive and constructive feedback. Below, we provide our response.
>
> One of the insights presented in our work is the role of rightward asymmetry stemming from the underlying Gumbel distribution, which contributes to variance reduction and is associated with a smoother loss landscape. This mechanism appears to be an important (though not the only) factor behind the consistent improvements observed across a wide range of tasks.
>
> To further investigate this property, in Appendix B, we applied this insight to Mish, which is based on a left-skewed distribution, and flipped its underlying distribution to construct Flipped Mish (FMish) with right-skewed asymmetry. Despite being a minimal modification, FMish significantly outperformed the original Mish, and also surpassed symmetric activations like Swish and GELU, though GoLU remained the top-performing activation. These results are consistent with our expectations based on the slope of each activation function: FMish has a slope of 0.4 at the origin, which is lower than that of Mish (0.6) and Swish/GELU (both 0.5), but slightly higher than GoLU (0.37).
>
> Based on our findings, the slope near the origin appears to be a meaningful control parameter that could serve as a guiding principle for designing future activation functions (as done for FMish), whether through theoretical analysis, automated methods, or by motivating parameterized activations (please also refer to the table on Asymmetry Ablation in response to reviewer AMoC).
>
> We hope these insights help illustrate the broader potential of our contribution. If there are any further questions or points that could help strengthen your view of the paper, we would be happy to continue the discussion at any time before the end of the discussion period on August 6th. Thank you.

---

> ### Comment · Reviewer_8T3r · 2025-08-09
>
> I would thank the authors for the response. I would like to keep my score and keep a positive view on the work.

---

> > ### Author Response · Authors · 2025-08-09
> >
> > We sincerely thank the reviewer for their constructive feedback and suggestions, and we greatly appreciate their decision to maintain a positive score.

---

### Official Review · Reviewer_bjVH · 2025-07-02

**Clarity:** 3
**Significance:** 2
**Originality:** 2
**Rating:** 4
**Confidence:** 4

**Summary:**

This paper introduces a new self-gated activation function, Gompertz Linear Units (GoLU), using the Gompertz function as the gate function. It demonstrates the variance reduction and smoothing in loss landscape properties of GoLU, and its consistent effectiveness on wide applications.

**Questions:**

1. Can the authors provide explicit notations of the functions in the captions of Figure 1 and Figure 2? Now, the textual description could be confusing and misleading (gate vs. gated functions).
2. Why only select three random images from ImageNet-1K for Figure 4? How about on a larger set of images or the whole ImageNet-1K? Similarly, it is better to include statistics at a larger scale for Table 1, e.g., more and larger images.
3. Is the variance reduction and smoothing effect consistent in language modeling? More results will be favorable and will provide stronger evidence.

**Ethical Concerns:**

["NO or VERY MINOR ethics concerns only"]

**Final Justification:**

The authors have provided valid responses to my questions and additional results that further support the effectiveness of their methods.

Given the solid analysis on the properties, valuable insights and promising results for future application shown in the paper, I would keep my borderline accept rating unchanged.

**Limitations:**

Yes

**Paper Formatting Concerns:**

I do not see any major formatting issues.

**Quality:**

3

**Strengths And Weaknesses:**

Strengths:
1. The paper provides insightful characterizations of the proposed GoLU, including its analytical form, the property to reduce variance based on curvature, and improvement on the smoothness of the loss landscape.
2. GoLU consistently outperforms other commonly used activation functions on wide applications (with acceptable exceptions), providing a new avenue for broader research.
3. Anonymized code facilitates reproducibility.

Weaknesses:
1. See my questions below.

---

> ### Author Rebuttal · Authors · 2025-07-30
>
> We sincerely thank the reviewer for their constructive feedback and questions. Below, we address the questions raised in the review.
>
> 1. We will update the captions of Figures 1 and 2 in the revised version of our paper to more clearly distinguish between the gated activations, their corresponding gate functions, and the distributions underlying those gate functions.
> 2. The three images in Figure 4 are randomly selected and are meant to visually showcase the variance reduction effect and support the theoretical argument. Including more samples would show qualitatively the same behavior and would therefore add limited value. We would also like to note that we observe such variance reduction effects across all intermediate layers of ResNet-50 trained on ImageNet-1k, not just in the final activation. We will comment on this in the revised version of the paper.
> 3. Motivated by the reviewer’s question, we selected babyGPT trained on TinyStories and conducted the same activation distribution analysis as in Figure 4. We observed that GoLU results in the lowest variance in the final layer’s activations: GoLU (Var: 0.042), GELU (Var: 0.066), ReLU (Var: 0.079), Swish (Var: 0.155).
> We further performed the same loss landscape analysis as in Figures 5 and 9, and again observed the same qualitative behavior, with GoLU exhibiting both the lowest mean loss and the lowest variance across the landscape.
> | Activation | Mean Loss     | Loss Variance     |
> |------------|-----------|--------------|
> | Swish      | 1.6301    | 3.94e-05     |
> | ReLU       | 1.6259    | 3.93e-05     |
> | GELU       | 1.6133    | 3.93e-05     |
> | GoLU       | **1.6079**    | **3.91e-05**     |
>
> We hope to have addressed the concerns raised by the reviewer. We respectfully ask that if you feel more positively about our paper, you kindly reconsider your rating accordingly. If not, please let us know if you have further questions or what can be further improved. We are happy to continue the discussion at any time before the end of the discussion period on August 6th. Thank you.

---

> > ### Comment · Reviewer_bjVH · 2025-08-01
> >
> > Thanks for the response.
> >
> > This work is easy to understand and shows promising results in most cases. I would suggest the authors validate its effectiveness across wide applications and domains (e.g, NLP, RL) due to its simplicity and integrate it into the popular framework. I assume it can work consistently well.

---

> > > ### Author Response · Authors · 2025-08-08
> > >
> > > We sincerely thank the reviewer for their constructive feedback and suggestions. We would like to highlight that the language modeling experiments in Section 3.3, as well as the machine translation results provided in Appendix I, offer evidence of GoLU's effectiveness on NLP tasks, though we agree this could be explored further.

---

### Official Review · Reviewer_qitQ · 2025-07-02

**Clarity:** 4
**Significance:** 1
**Originality:** 2
**Rating:** 5
**Confidence:** 4

**Summary:**

The paper introduces a novel activation function, Gompertz Linear Unit (GoLU). The authors present GoLU as a self-gated activation that reduces variance in latent representations and promotes smoother gradient flows.
The experimental results show improved performance across various tasks such as image classification, language modeling, semantic segmentation, object detection, etc.

**Questions:**

Q1. Could the authors offer deeper theoretical analysis or empirical evidence to rigorously justify the claimed advantages of rightward asymmetry in GoLU?

Q2. Do the performance gains transfer with advanced training procedures as in [1] mentioned in W1? Further, can GoLU seamlessly replace ReLU in pretrained models and lead to better performance in a fine-tuning setting on a downstream task?

If Q2. is answered I could raise my score, because as the paper's contribution relies on the experimental part, I would like to see a significant gain before granting acceptance.

**Ethical Concerns:**

["NO or VERY MINOR ethics concerns only"]

**Final Justification:**

See the comment "answer to rebuttal".

**Limitations:**

yes

**Quality:**

2

**Strengths And Weaknesses:**

**Strengths:**

- S1 : Well written paper that is easy to follow. The introduction of GoLU is clear and straightforward to understand.

- S2 : The amount of experiments is commendable, with a comprehensive empirical evaluation on diverse benchmark datasets. As this paper contribution is empirical, this is a good points (however see W1).

- S3 : The code is available and a Cuda kernel was developed. This allows for the community a seamless integration and use of GoLU.

---

**Weaknesses:**

- W1 : The improvements in performance, while seamingly widespread, appear (very) modest in magnitude, raising questions about their real-world impact. Furthermore, as different well-known training procedures would lead to better baselines (with even better results than those reported for GoLU), I am not convinced by the usefulness of this new activation function. I give a couple of examples:

    a) Table 2 and 3 baselines are a bit questionable.

    - for table 2, the results and training procedure provided by [1] with even the cheapest procedure (A.3) lead for a ResNet50 on Imagenet to 78.1 % while the paper reports 75.44% for ReLU and 76.63% for GoLU.

    - for table 3, as public implementations, for example [this git](https://github.com/akamaster/pytorch_resnet_cifar10) can lead to much better results with ReLU. I am a bit worried about the reported gains, see the table below:
| Name       | Test err (original impl.) | Test err (public git impl.) | Test err (reported for ReLU) | Test err (reported for GoLU) |
|------------|------------------|------------------------|------------------|------------------|
| ResNet20   | 8.75            | 8.27                  | 8.59        | 8.23        |
| ResNet32   | 7.51            | 7.37                  | 7.79        | 7.31        |
| ResNet44   | 7.17            | 6.90                  | 7.42        | 7.15        |
| ResNet56   | 6.97            | 6.61                  | 7.20        | 6.85        |
| ResNet110  | 6.43            | 6.32                  | 6.79       | 6.75        |

   b) The Perplexity gains reported in Table 4 are in my opinion, not significant enough to be convincing and one could wonder whether when applied on larger language models, GoLU can lead to better results.

- W2 : The discussion on smooth/flat landscape is not an argument.
Figure 5, meant to demonstrate smoother loss landscapes, lacks clarity and fails to convincingly support the authors' claims, regarding that point (projecting a ~250k dimensional space onto a 3 dimensional one is not very interpretable). Further, as discussed in [2], these metrics are to be handled with extra attention and not just given as arguments.

- W3 : The theoretical contribution is limited, as the paper primarily offers incremental enhancements over existing methods based on an established mathematical function, rather than presenting fundamentally novel theoretical insights.

[1] Wightman, R., Touvron, H., & Jégou, H. (2021). Resnet strikes back: An improved training procedure in timm. arXiv preprint arXiv:2110.00476.

[2] Andriushchenko, M., Croce, F., Müller, M., Hein, M., & Flammarion, N. (2023). A modern look at the relationship between sharpness and generalization. arXiv preprint arXiv:2302.07011.

---

> ### Author Rebuttal · Authors · 2025-07-30
>
> We sincerely thank the reviewer for their constructive feedback and questions. Below, we address the points raised in the review.
>
> **Q1 and W3 (theoretical insight and empirical evidence):**
> We would like to emphasize that, to the best of our knowledge, this is the first work to introduce asymmetry as a novel and relevant concept in the design of activation functions. This contribution is not merely an empirical observation. We provide a theoretical analysis that directly links rightward asymmetry to variance reduction in latent representations. Specifically, we show that the reduced slope around the origin, a result of the right-skewed asymmetry of the underlying Gumbel distribution, acts as a form of implicit regularization, which is desirable in deep learning.
>
> Beyond the consistently strong performance of GoLU across a wide range of tasks, including vision, language modeling, and generative modeling, Appendix B provides additional empirical evidence highlighting the advantages of rightward asymmetry. We applied our insights to Mish, which is based on a left-skewed distribution, and flipped its underlying distribution to construct Flipped Mish (FMish) with right-skewed asymmetry.
>
> Despite being a minimal modification, FMish significantly outperformed the original Mish, and also surpassed symmetric activations like Swish and GELU when evaluated on ResNet18, ResNet50, and ViT-B/32 trained on ImageNet-1k, though GoLU remained the top-performing activation. This trend aligns closely with our theoretical analysis regarding slope and variance control:
> - Mish: slope = 0.6 at the origin
> - Swish / GELU: slope = 0.5 at the origin
> - FMish: slope = 0.4 at the origin
> - GoLU: slope ≈ 0.37 at the origin
>
> **Q2 and W1 (Transfer gains with advanced training procedures):**
> While we understand the reviewer’s concern that some of the performance gains may appear small, we respectfully emphasize that even fractional improvements in accuracy can lead to substantial impact when amortized across many different deep learning applications. Activation functions like GELU and Swish also showed small improvements over ReLU, but they are now widely adopted, improving results in thousands of code bases. We specifically emphasize that, unlike many activations that show task or architecture-specific benefits, GoLU provides more consistent improvements across domains, including vision, language modeling, and generative modeling, as demonstrated in our experiments.
>
> We would also like to emphasize a trend we observed in our Table 2 that the difference in top-1 accuracy between GoLU and GELU on ImageNet increases for larger models:
> - RN18: +0.10
> - RN34: +0.27
> - RN50: +0.56
> - WRN50-2: +0.65
>
> Regarding the concern about baseline performance and training pipelines, we fully agree with the reviewer that modern training recipes can significantly improve baseline results. However, our goal in this paper is not to surpass SOTA, but rather to demonstrate that GoLU provides consistent benefits when substituted into established pipelines, without further tuning. For this reason, we standardized our training setup across activations (focusing primarily on original implementations when possible) to ensure a fair and controlled comparison.
>
> That said, based on the reviewer’s suggestion, we trained ResNet-50 on ImageNet-1k using the A3 training recipe from [1]. We used the timm library and followed the hyperparameters listed in Table 2 of [1]. With three random seeds (0,1,2), the ReLU model achieved a mean top-1 validation accuracy of $\bf{78.17  \pm 0.039}$, consistent with [1]. We then simply retrained the model with GoLU, keeping everything else identical, and observed a mean top-1 validation accuracy of $\bf{79.15 \pm 0.024}$, which is a highly significant improvement over the default ReLU activation. This improvement is also consistently observed across the entire training curve.
>
> Regarding the question on fine-tuning, while fine-tuning a ReLU-pretrained model using GoLU may yield improvements, we have not verified this, and this is not a claim made in our paper. It is conceivable that the weights of a model pretrained with ReLU are already aligned with that activation, and simply replacing it during fine-tuning may not reveal the full strength or effect of GoLU. Instead, we expect that future foundation models pretrained with GoLU will show improved performance, which will in turn benefit the resulting fine-tuned variants.
>
> [1] Wightman, R., Touvron, H., & Jégou, H. (2021). Resnet strikes back: An improved training procedure in timm. arXiv preprint arXiv:2110.00476.
>
> **W2 (discussion on smooth loss landscape):**
> As noted in the manuscript, the smoother loss landscape observed with GoLU is an empirical observation (rather than a formal argument). While it aligns with properties like GoLU’s smaller slope and variance-reducing effect, it is meant to complement the main experimental results.
>
> We agree that loss landscape visualizations must be interpreted cautiously. However, we would like to clarify that Figure 5 simply illustrates a smoother loss landscape along a 2D subspace spanned by random directions, in order to provide qualitative insight. While not definitive, this is consistent with GoLU’s observed generalization gains. We will clarify this in the final version to avoid potential over-interpretation.
>
> We hope to have addressed the concerns raised by the reviewer. We respectfully ask that if you feel more positively about our paper, you kindly reconsider your rating accordingly. If not, please let us know if you have further questions or what can be further improved. We are happy to continue the discussion at any time before the end of the discussion period on August 6th. Thank you.

---

> > ### Comment · Reviewer_qitQ · 2025-08-01
> > **Answer to rebuttal**
> >
> > First of all, I would like to thank the authors for their detailed rebuttal.
> > I have to admit that the reject score was primarily based on the *weak improvements* over the baselines especially when they were not set particularly high. The new results with carefully chosen hyper-parameters are much more convincing.
> > I will update my score to an accept (5) accordingly.
> >
> > - **Q1/W3**:
> >
> > I agree with the authors' rebuttal that the paper actually does present *some* theoretical insights that I misjudged to some extent. Although more exploration on this subject could be very interesting, the presented analysis can be considered as satisfying.
> >
> > - **Q2 and W1**
> >
> > As said in the introduction of this comment, I am very pleased to see that GoLU leads to increased performance even with recipes adapted to other activation functions. The added results are very convincing in my opinion.
> >
> > > "Instead, we expect that future foundation models pretrained with GoLU will show improved performance, which will in turn benefit the resulting fine-tuned variants."
> >
> > I agree with the authors, however the cost of training ever-bigger new foundation models might be undesirable. It would be very interesting to see if current foundation models can be "distilled" in a GoLU version and and analyze the performance transfer (if there is one).
> >
> > - **W2: About the smooth landscape**
> >
> > A nice (although maybe tedious to obtain) addition for the smoothness argument might be a theoretical derivation in small dimension. I am never very convinced about smoothness arguments in high dimension as those spaces are very hard to apprehend. However I appreciate the authors' effort to clarify their point in the new version to avoid over-interpretation of their observation.

---

> > > ### Author Response · Authors · 2025-08-08
> > >
> > > We sincerely thank the reviewer for their constructive feedback and suggestions. We greatly appreciate their decision to reassess the paper and raise their score.

---

### Official Review · Reviewer_g3hi · 2025-07-03

**Clarity:** 3
**Significance:** 3
**Originality:** 3
**Rating:** 5
**Confidence:** 4

**Summary:**

The authors propose Gompertz Linear Units (GoLU), a novel activation function based on the Gompertz function. The paper formally defines GoLU and analyzes its key properties. In particular, the authors discuss its role during training, highlighting that it induces a smoother loss landscape and a more spread-out weight distribution, they are considered better compared to previous activation functions. Experimental results show that the proposed method achieves better performance across a wide range of tasks, including but not limited to image classification, object detection, image generation, and language modeling.

**Questions:**

It should benefit the readability and clarity of the paper if some of the theoretical parts in the appendix are included in the main body.

**Ethical Concerns:**

["NO or VERY MINOR ethics concerns only"]

**Quality:**

4

**Strengths And Weaknesses:**

Pros
1.	This paper is well-written.
2.	Significant improvements have been achieved in many commonly used benchmarks.
3.	In addition to strong baseline performance, the paper includes numerous visual comparisons (including those in the appendix) that help illustrate the advantages of the proposed method.
4.	Anonymous code repo is provided, supporting reproducibility.
Cons
1.	There could be more visualization results, such as Table 1, and perhaps an ImageNet one, which would be more convincing.
2.	The proposed method shows impressive performance improvements. However, the main text lacks theoretical explanations that help readers understand why the method works. Also, I noticed that the appendix contains some important analysis, which would strengthen the paper if included in the main body

---

> ### Author Rebuttal · Authors · 2025-07-30
>
> We sincerely thank the reviewer for their positive and constructive feedback. Below, we provide our response.
>
> **Con 1:**
> We have indeed conducted an ImageNet equivalent of the experiment in Table 1 and observed a qualitatively similar variance reduction effect. However, we chose to include the CIFAR-10 results because, to the best of our knowledge, ImageNet images cannot be shared publicly due to license restrictions. In the revised version of the paper we will explicitly mention that the same effect is observed on ImageNet.
> Additionally, Figure 4 offers a similar analysis on randomly sampled ImageNet images but shows the full activation density distributions. We would also like to highlight that this variance reduction effect is not limited to the final activation, as shown in Figure 4, but is consistently observed across intermediate layers of ResNet-50 as well. We will add a note about this in the updated version.
>
> **Con 2 & Question:**
> Due to space constraints, we had to place several interesting results in the appendix. If the manuscript is accepted, we will have an additional page to move some of the more important results from the appendix, including some theoretical arguments, into the main body.
>
> We hope that our responses have addressed the concerns raised by the reviewer. If there are any further questions or points that could help strengthen your view of the paper, we would be happy to continue the discussion at any time before the end of the discussion period on August 6th. Thank you.

---

### Note · Authors · 2025-08-15

We would like to once again thank all the reviewers for their time, thoughtful feedback, and positive evaluations. We hope our rebuttal has satisfactorily addressed the questions and concerns raised during the review process.

We would like to take this opportunity to emphasize that our work introduces new insights into activation function design, supported by theoretical arguments, particularly highlighting the role of asymmetry and its implications. Building on these insights, we propose a novel activation function derived from the Gumbel distribution, which consistently outperforms standard baselines across a wide range of diverse tasks in extensive experiments. We further provide a custom CUDA kernel implementation that makes GoLU efficient and scalable in practice.

Given the ubiquity of activation functions in neural networks, we believe GoLU has the potential to be broadly adopted by the deep learning community and to make a significant impact.

---

### Decision · Program_Chairs · 2025-09-17

**Decision:**

Accept (poster)

**Comment:**

The paper introduces a new activation function, Gompertz Linear Unit (GoLU): $x e^{-e^{-x}}$. The authors claim that GoLU as a self-gated activation, which is able to reduce variance in latent representations and promotes smoother gradient flows.
The comprehensive experimental results on image classification, language modeling, semantic segmentation, object detection, support the effectiveness of GoLU.

Extensive experiments across multiple tasks—including image classification, language modeling, semantic segmentation, and object detection—demonstrate the effectiveness of GoLU.

During the review process, several reviewers raised concerns regarding the consistency of results, the extent of performance improvements, and the role of hyperparameter tuning. In response, the authors provided additional experiments in the rebuttal, which successfully addressed these concerns and convinced all reviewers.

The AC, while acknowledging the positive outcome, noted that most of the presented experiments are image-centric (e.g., classification, segmentation, detection). To further **strengthen the significance and impact** of the work, the AC suggested that the authors evaluate GoLU on large language models (e.g., a 1B-parameter model).